# Neural computations underlying strategic social decision-making in groups

Seongmin A. Park [1,2]*, Mariateresa Sestito[1], Erie D. Boorman[2] & Jean-Claude Dreher[1]*

When making decisions in groups, the outcome of one's decision often depends on the decisions of others, and there is a tradeoff between short-term incentives for an individual and long-term incentives for the groups. Yet, little is known about the neurocomputational mechanisms at play when weighing different utilities during repeated social interactions. Here, using model-based fMRI and Public-good-games, we find that the ventromedial prefrontal cortex encodes immediate expected rewards as individual utility while the lateral frontopolar cortex encodes group utility (i.e., pending rewards of alternative strategies beneficial for the group). When it is required to change one's strategy, these brain regions exhibited changes in functional interactions with brain regions engaged in switching strategies. Moreover, the anterior cingulate cortex and the temporoparietal junction updated beliefs about the decision of others during interactions. Together, our findings provide a neurocomputational account of how the brain dynamically computes effective strategies to make adaptive collective decisions.

---

[1] Neuroeconomics laboratory, Institut des Sciences Cognitives Marc Jeannerod, CNRS UMR 5229, 69675 Lyon, France. [2] Center for Mind & Brain and Department of Psychology, University of California Davis, Davis, CA 95618, USA. *email: seongmin.a.park@gmail.com; dreher@isc.cnrs.fr

When decisions are made in a social context, the degree of uncertainty about the possible outcomes of our choices depends upon the decision of others. This makes it extremely challenging to study the nature of the computational algorithms used by the brain that can account for human social decision-making. During collective decisions in groups, individuals often make decisions on whether or not to contribute their resources to public goods, which is launched if and only if a certain level of contribution is reached. Examples of such collective decisions include recycling resources, casting a ballot, and reviewing grant proposals. In such collective decisions, volunteers improve the welfare of other individuals[1]. However, their contribution is wasted if there are too many volunteers. Yet, the public project fails if not enough volunteers contribute. This social dilemma is called the volunteer's dilemma[2–4]. In such volunteer's dilemma, utility of one's decision depends on the decision of others. When such collective decisions are made repeatedly within the same group, it is therefore crucial for the brain to track and update one's belief about the decision of others at each social interaction. In particular, the brain needs to compute not only how much more benefit can be expected from the immediate interaction in choosing one alternative over another but also how much more the group can benefit from the collective rewards from the remaining interactions. The brain needs to weigh these individual and group utilities to select the optimal strategy to maximize one's total benefits in social interactions. Despite the ubiquity of collective decision-making in society, it is still unknown how the brain computes individual and group utilities and makes such strategic decisions during collective decision-making.

One important question faced by decision neuroscience in the context of repetitive social interactions is to know whether the brain computes the utility of different strategies according to the belief of the decision of others in separate or identical regions, and further, how these computations are used by neural circuits implementing the strategic decision. To investigate these questions, we used model-based fMRI and a threshold public goods game (PGG)[5–7]. During a threshold PGG, individuals make decisions with the same group members repeatedly for a finite number of social interactions. The group obtains the rewards only when more than a specific number of members contribute their resources. Such a rule induces individuals to make strategic decisions about when to contribute their resources and when to free-ride. This means that each group member assigns specific probabilities to multiple pure strategies for making a decision, and that the optimal decision varies dynamically according to one's belief about the decision of others[7–10]. Previous fMRI studies have identified a neural network that is engaged in learning and tracking the mental state of the counterpart during dyadic interactions[11–15]. However, the neural mechanisms underlying repeated social interactions when it comes to inferring the intentions of others in groups of more than two individuals, to compute immediate and long-term utility of different strategies and select a strategy, remain unaddressed.

In the volunteer's dilemma, there is a tradeoff between immediate incentives and long-term collective rewards. Choosing to free-ride in the current interaction can give a larger immediate reward for an individual while it could also decrease the expected collective rewards for the group in a longer-term by making the others less cooperative. We tested different computational models to investigate how the signal updating our beliefs about the intentions of others is used to determine the long-term group utility and weigh it against the short-term individual utility in order to make strategic decisions. Our results show that the anterior cingulate gyrus (ACCg) and temporoparietal junction (TPJ) track one's belief about the likely decision of others.

Moreover, we identify subdivisions of the prefrontal cortex which computed immediate individual utility and long-term collective utility for the group, respectively, in the ventromedial (vmPFC) versus lateral frontopolar cortex (lFPC). Finally, when participants can expect better utility by choosing the alternative strategy, the ACC and the ventrolateral prefrontal cortex (vlPFC), which are engaged for switching decision strategy, show changes in functional connectivity with the vmPFC and lFPC, regions encoding the utility of the strategy. Together, these results provide a model-based account of the neurocomputational mechanisms guiding human strategic decisions during collective decisions.

## Results

**Public goods game.** Participants played a series of threshold PGG[5–7] in an MRI scanner. Each of the 12 PGGs comprised a finite number of interactions ($T = 15$). On each trial, individuals in a five-member group made a private decision to either contribute their endowed resource to the group or to free-ride by keeping it. The contribution was not returned to them. The benefits were equally allocated to all participants if the public goods were produced (Fig. 1a). In this study, the public goods were only generated when the contribution of the group was $k$ or more than $k$. Therefore, $k$ is the minimum amount to generate the public goods. Participants had been informed about the meaning of $k$ and about the number of trials during which they will be playing with the same partners.

We modulated the level of the volunteer's dilemma by changing the threshold $k$ of this minimum number of contributors required to generate the public goods ($k = 2$ or $k = 4$). When two or more contributions are required to benefit from the public goods ($k = 2$), one gets the group reward (2 MU) when the number of free-riders $nF \leq 3$, while her contribution is not serving to generate the public goods (others' contribution is enough to generate the public goods), but wasted when $nF \leq 2$. During the PGG in which four or more contributions are required to benefit from the public goods ($k = 4$), one gets the group rewards when $nF \leq 1$, but her contribution is wasted if all contributes. Adopting the threshold PGG allows us to investigate strategic decision-making when confronting the volunteer's dilemma. Specifically, the probability that one's contribution is wasted is higher when fewer contributors are required to generate public goods, which incurs a stronger volunteer's dilemma ($k = 2$, stronger volunteer's dilemma; $k = 4$, weaker volunteer's dilemma). To control the underlying motivations of other individuals across participants while creating plausible behavior in social interactions, decisions of other members of the group were generated by a computer program, unbeknownst to the participants. Indeed, in neuroimaging studies, it is critical that the brain responses of all participants are modulated by a specific range of controlled computational variables. If the behavior is uncontrolled but is acquired while facing real humans, then there is a very high chance that the participants could interact with individuals that have different motives underlying their decisions (see decisions of other members of the group in the Methods section). Details of the PGG describing VD and its payoff matrix are described in the Methods section (Fig. 1b).

**Behavioral results.** First, we examined whether participants changed their contribution rate under different levels of volunteer's dilemma. When public goods are produced with a larger number of contributors ($k = 4$) compared with a smaller number of contributors ($k = 2$), participants tend to allocate their resources to the group more ($t_{24} = 5.01$, $p < 0.01$, paired $t$ test). A tendency to make more contribution under weak volunteer's dilemma was found across trials. That is, participants were more likely to contribute their resources during the PGG when confronted with the weak

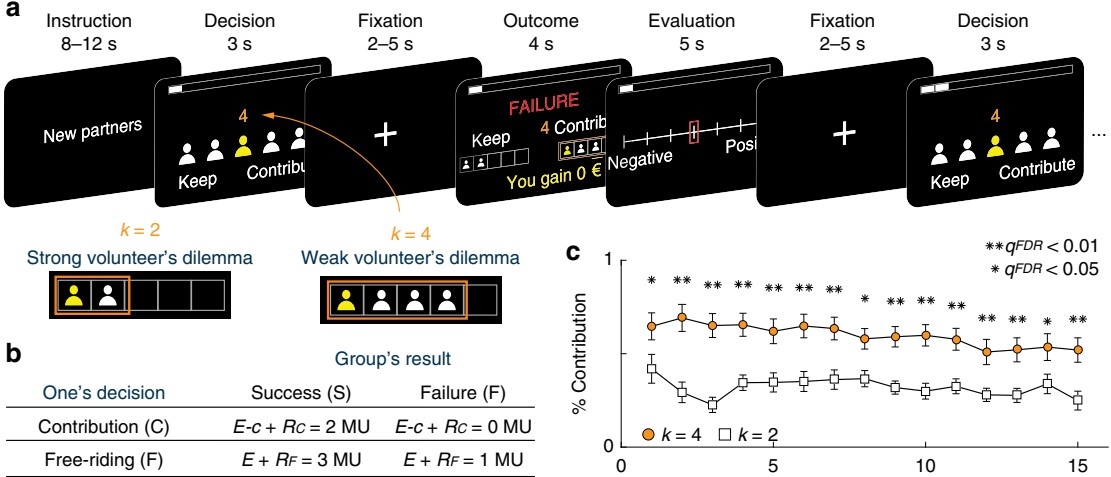

**Fig. 1** Task design and behavioral results. **a** Representative trial and timeline of the public goods game (PGG). The level of volunteer's dilemma is modulated by the decision threshold, $k$ ($k = 2$ or $k = 4$), which is the minimal number of contributors required to produce public goods. When the outcome is revealed, the choice of the participant is indicated by the yellow person, and the choices of others are indicated by the white persons. The orange rectangle represents the decision threshold $k$ needed to implement the PGG. **b** The payoff matrix of PGG. The payoffs depend on the participant's decision and on whether the mutual contribution exceeds the stated minimum ($k$) or not. On each trial, participants were endowed 1 monetary unit ($E = 1$ MU). They made a binary decision between contribution and free-riding. The cost of contribution was 1 MU ($c$). The group reward is $R_C$ when a participant contributes, and $R_F$ when the participant free-rides. Every member of the group gets the reward ($R = 2$ MU) when the group succeeds to generate public goods, and the group reward is 0 MU when the group fails to generate public goods. **c** Average probability to contribute in each round. The weaker the volunteer's dilemma ($k = 4$) the more contribution decisions participants made during PGG. The contribution rate decayed faster when confronted to a weaker ($k = 4$) compare to a stronger ($k = 2$) volunteer's dilemma. Error bars indicate s.e.m. which are equivalent throughout the figure; **$q < 0.01$, *$q < 0.05$ FDR corrected for multiple comparisons.

volunteer's dilemma ($k = 4$) than when experiencing the stronger volunteer's dilemma ($k = 2$) (false discovery rate (FDR)[16] corrected for multiple comparison; Fig. 1c). That is, the stronger the volunteer's dilemma, the less participants were likely to contribute and the more they relied on others' contributions. These findings suggest that participants adopted a mixed strategy in a repetitive threshold PGG (Supplementary Eq. (1)).

**Model-free analysis**. Using a mixed-effect logistic regression model, we examined how participants' contribution decision at a given round, $t$ was influenced by relevant variables, such as the number of free-riders ($nF$), the previous decision ($D$), the success or failure ($S/F$) to generate the public goods, and the win-stay and loose-switch strategy ($Ws/Ls$). We also included these regressors up to $t$-3 previous trials. We found that participants were neither contributing more when the number of other free-riders increased ($t_{24} = 1.73$, $p = 0.10$, one-sample $t$ test) nor when there was success in generating the public goods ($t_{24} = -0.30$, $p = 0.77$, one-sample $t$ test). This result suggests that participants generally used a model to make a strategic decision rather than simply repeating their decisions that generated the public goods in previous interactions in a model-free way (Fig. 2a).

**Computational models**. To elucidate the computations that participants employ when faced with the volunteer's dilemma, and to generate quantitative predictions of our fMRI data, we compared four competing computational models. First, the social-learning model (SL) assumed that participants compute the immediate expected reward allocated to the individual and the long-term collective utility allocated to the group of their decision separately. Specifically, the probability that a participant makes a contribution to the group on trial $t$ depends on the decision value $Q$, representing the weighted sum of the immediate utility for individuals, $I_t$ and the long-term collective utility for groups, $G_t$ (Eq. (1)). The $I_t$ reflects the relative reward expected when contributing as compared with when making a free-riding decision on the current trial,

$t$ (Eq. (2)). We adopted the model proposed by Archetti et al.[4] that also integrates the altruistic rewards given to the other members (Eq. (4)) into the computation of one's expected utility in order to account for the possibility that subjects derived additional value (positive or negative) from others' rewards (Eq. (5)). Participants in PGG with finite repetitions with the same partners can expect larger rewards in the long-term if their group is cooperative. The group utility, $G_t$ represents the cumulative expected rewards of the public goods that a player expects for remaining interactions, as proposed by Wunder et al.[17] (Eq. (8)). Both of the expected utilities ($I_t$ and $G_t$) varied according to their belief about the decision of others ($\gamma_t$). Therefore, this model incorporates the fact that participants continuously predict the upcoming decision of others by learning from the previous interactions (Eq. (10)).

Second, the myopic model assumed that participants only take individual utility into account (Eq. (13)). Compared with the social learning model that incorporates the importance of future expected utility, the agent of the myopic model only considers immediate rewards when making decisions. Third, the forward-looking model assumed that participants only take into account long-term collective utility for groups (Eq. (14)). Fourth, the inequity aversion model assumed that participants contribute with a certain probability until their group benefits from the public goods (Eq. (15)). Specifically, the inequity aversion model predicts that participants are more likely to change their contribution decisions to free-riding when they experience inequity in their contribution to the group.

**Model validation and comparison**. To test whether the social learning model captures the characteristics of decisions during PGG, we performed a number of analyses. First, we fitted those four computational models to the participants' actual choice data. Using the Bayesian information criteria (BIC; Eq. (16)) which penalizes additional free parameters, we compared the goodness of fit of each model (Table 1). We found that the social learning model better explained participants' decision during PGG than other alternative

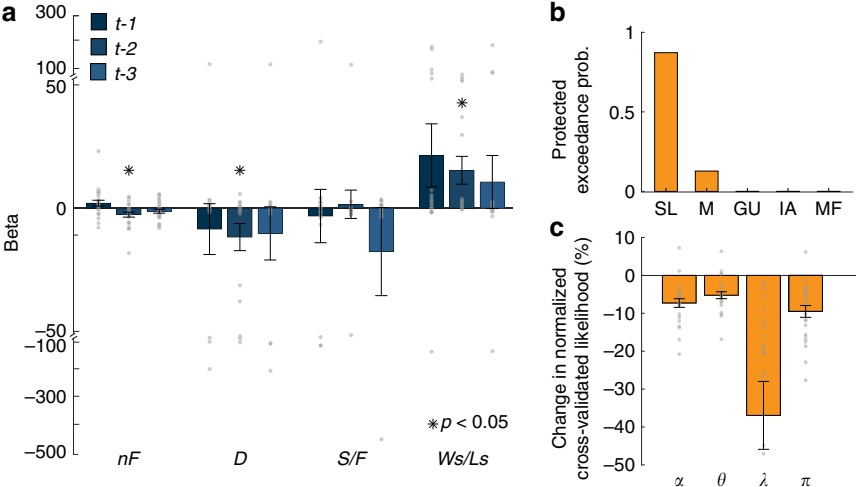

**Fig. 2** Model validation and comparison. **a** Model-free analysis. We regressed the behavioral decision on the number of free-riders (*nF*), decision (*D*), success or failure to generate the public goods (*S/F*), and win-stay and lose-switch strategy (*Ws/Ls*) in previous trials up to three trials back. *$p < 0.05$ (mixed-effect logistic regression). **b** Model comparisons based on the Bayesian model selection. The protected exceedance probabilities indicate that the social learning (SL) model explains decisions in PGG better than other alternative models: myopic (M); group utility (GU); inequity aversion (IA); model-free analysis (MF). **c** Changes in quality of fit resulting from removing free parameters in the social learning model; $\alpha$, learning rate; $\theta$, weight on learning rate; $\lambda$, contribution cost; $\pi$, willingness to make altruistic decisions.

**Table 1 Parameter estimates and model comparison of behavior.**

| Models | | | | | | BIC |
|---|---|---|---|---|---|---|
| Social learning | $\omega$<br>0.65 ± 0.07 | $\pi$<br>0.16 ± 0.16 | $\lambda$<br>−2.10 ±0 .45 | $\alpha$<br>0.51 ± 0.06 | $\theta$<br>0.13 ± 0.02 | −6149 |
| Myopic | $\omega$<br>11.04 ± 2.76 | $\pi$<br>0.34 ± 0.19 | $\lambda$<br>−2.98 ± 1.09 | $\alpha$<br>0.59 ± 0.06 | $\theta$<br>0.14 ± 0.04 | −5721 |
| Group utility | $\chi$<br>9.30 ± 4.55 | $\zeta$<br>−30.91 ± 4.92 | $\alpha$<br>0.56 ± 0.06 | $\theta$<br>0.12 ± 0.02 | | −5713 |
| Inequity aversion | $\delta$<br>16.39 ± 7.11 | $\varepsilon$<br>0.61 ± 0.42 | $\kappa$<br>1.37 ± 1.00 | | | −4220 |

The parameter estimates in social learning, myopic, group utility, and inequity aversion models (mean across participants ± SEM) and a comparison of these models, including the Bayesian Information Criterion (BIC). The social learning (SL) and the myopic (M) models have five parameters, respectively: $\omega$ is the weight assigned to the individual utility ($1 − \omega$ is, therefore, the weight assigned to the group utility in the social learning model); $\pi$ denotes the tendency to make an altruistic contribution; $\lambda$ denotes the subjective cost of contribution; $\alpha$ denotes the learning rate of social prediction error; $\theta$ denotes the weighting term of the reward prediction error on the learning late; the group utility (GU) model has four parameters: $\chi$ is the weight assigned to the group utility; $\zeta$ is the initial bias (error term); $\alpha$ denotes the learning rate of social prediction error; $\theta$ denotes the weighting term of the reward prediction error on the learning late. The inequity aversion (IA) model has three parameters: $\delta$ denotes the weight assigned to the level of inequity between one's mean contribution and that of others; $\varepsilon$ denotes the sensitivity to group rewards; and $\kappa$ indicates the noise in decision process

models, and this was also true when comparing the posterior model probabilities using Bayesian model selection (BMS, Fig. 2b).

Second, we conducted a leave-one-block-out cross-validation approach such that the decision made for the N-th PGG is predicted based on the parameters estimated by fitting the model to decisions made for the other 11 PGGs (see Supplementary Methods and Supplementary Fig. 1). Last, we tested the contribution of free parameters of social learning models to the quality of the fit (see Supplementary Methods and Fig. 2c).

Taken together, these analyses show that behavior in the volunteer's dilemma can be best captured by the social learning model. According to this model, people compute the following key variables: individual utility, group utility, their integration, a prediction of the group's likely choice, and a corresponding social prediction error. Next, we harnessed quantitative predictions from the social learning model to identify the neural correlates of these computations.

**Testing different levels of iterative reasoning**. To address the influences of iterative reasoning on decision-making during the

PGG[11,18,19], we tested alternative social learning models in which the individual utility and the group utility are updated based on one's belief using 2nd and 3rd order belief reasoning (Eq. (11)). We further compared their predictabilities of contribution decisions with that of the 1st order belief model. We found that higher-order reasoning did not explain the decisions better than the 1st order belief model (social learning model) (Supplementary Fig. 2). The fact that participants, in this study, were less likely to use higher-order reasoning may be due to the feedback provided, to the finite number of interactions with the same partners and/or to the fact that we did not explicitly show the decision of each player but the proportion of contributors in the group. Notably, this setup mimics many ecologically relevant group decision-making situations.

**Neural encoding of individual utility and group utility**. First, we analyzed the fMRI data to search for brain regions tracking key computational variables identified in the behavioral analyses. Using a general linear model, we modeled brain responses at the outcome phase on trial $t − 1$ to predict the next decision on trial $t$ (GLM1). The BOLD signal in the ventromedial prefrontal cortex

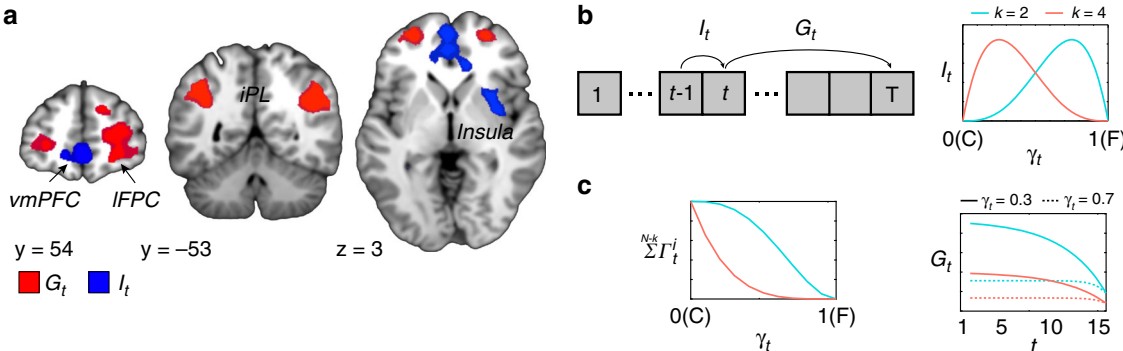

**Fig. 3** Neural correlates of 'Individual utility' and 'Group utility' from the social learning model. **a** Activity (in blue) in the ventromedial prefrontal cortex (vmPFC) at the time of feedback on trial $t-1$ inversely correlated with individual utility ($I_t$) which indicates the expected utility of contribution decision compared with the expected utility of free-riding on the current trial. Activities (in red) in the right lateral frontopolar cortex (lFPC) and bilateral inferior parietal lobule (IPL) at the time of feedback on trial $t-1$ positively correlated with the estimated group utility ($G_t$), which indicates to what extent a participant expects long-term rewards from the interactions with the group for the remaining interactions. Those areas are thresholded at $p < 0.05$, FWE corrected at the cluster level. For illustrative purposes, the statistical maps are thresholded at $p < 0.005$, uncorrected (darker color). The lighter color map was thresholded at $p < 0.001$, uncorrected. **b** After the feedback of the previous trial, $t-1$, a participant may compute the immediate expected utility for oneself ($I_t$) as well as compute the cumulative expected rewards for the group ($G_t$) for remaining interactions ($T-t+1$, where $T=15$). The right panel is the conceptual illustration of individual utility ($I_t$) according to one's belief about the decision of another ($\gamma$). It correlates with the binomial probability density function of $\gamma_t$, which indicates one's belief about the probability that one of the other members will free-ride at round $t$. **c** Conceptual illustration of group utility ($G_t$) according to one's belief about the decision of another ($\gamma$). $G_t$ depends on the probability that the group generates the public goods and the number of remaining trials. (Left) The probability to generate the public goods varies as a binomial cumulative density function given $\gamma_t$ at trial $t$. (Right) Relationship between $G_t$ and $\gamma_t$. $G_t$ is high when participants believe that another player is less likely to free-ride (e.g., $\gamma_t = 0.3$) compared with when they believe that another player is more likely to free-ride (e.g., $\gamma_t = 0.7$, dotted line). $G_t$ is also discounted by the number of remaining interactions.

(vmPFC), $(x, y, z) = (0, 56, -2$, the peak voxel in MNI coordinates) and anterior insula $(x, y, z) = (36, -1, 7)$ were inversely correlated with $I_t$ (Fig. 3a; Supplementary Table 1, one-sample $t$ test throughout fMRI analysis, $p < 0.05$ whole-brain family-wise error (FWE) corrected at the cluster level). Since $I_t$ was computed as the utility of one's contribution compared with the expected reward from a free-riding decision, this inverse relationship indicates that the vmPFC activity encodes the expected immediate reward of choosing to free-ride relative to that of contribution in the trial $t$ (Fig. 3b).

Having identified the brain regions computing $I_t$, we searched for the brain systems computing $G_t$, the other key computation guiding decisions implied by the social learning model. Activity in the lateral frontopolar cortex (lFPC; $x, y, z = 24, 50, -8$), precuneus $(x, y, z = -6, -67, 43)$, and the bilateral inferior parietal lobule (IPL; left iPL $x, y, z = -39, -58, 37$; right IPL $= 45, -52, 34$) encoded group utility, $G_t$, at the time of outcome (Fig. 3a; Supplementary Table 1, $p < 0.05$ whole-brain FWE corrected at the cluster level; GLM1). Since $G_t$ reflects the pending rewards for remaining interactions ($T-t+1$, with $T=15$), estimated by the probability that the group generates public goods at trial $t$, higher activity in this brain system reflects a greater chance of switching their future decision to contribution by pursuing greater cumulative rewards in the long-term (Fig. 3c).

Previous studies indicate that during repeated social interactions in groups, individuals are more likely to update their belief when observing the decision of others at trial $t-1$, which explains variance in decision-making at trial $t$[20–22]. For this reason, we modeled the brain responses at the outcome phase— i.e., the utility of the decision at trial $t$ was modeled at the time of receiving the outcome of social interactions at trial $t-1$. To test the alternative hypothesis that the computational variables are encoded at the time of decision-making, we analyzed the fMRI data at the time of the decision-making phase. The brain responses were modeled in the same way as in GLM1, except that we modeled brain responses at the decision phase on trial $t$. We found that activity in the vmPFC $(x, y, z = 0, 59, 1)$ was inversely correlated with the model estimated $I_t$, and lFPC $(x, y, z = 39, 44,$

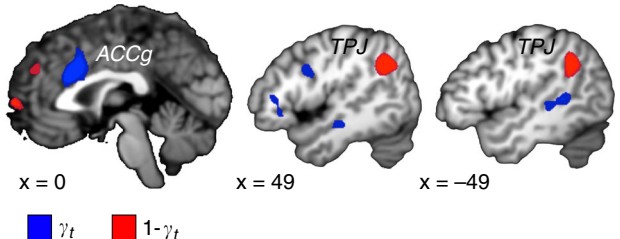

**Fig. 4** Brain areas tracking the belief about the decision of another member in the group. Brain areas updating the belief about the decision of another member in the group when the outcome of interactions is revealed. Activity in the anterior cingulate gyrus (ACCg) correlated with model estimates of the belief $\gamma_t$ of participants about the extent to which one of the other members in the group free-rides in the current trial (blue). Activity in the bilateral temporoparietal junction (TPJ) correlated with model estimates of the belief $(1-\gamma_t)$ of participants about the extent to which one of the other members in the group contributes at the current trial, $t$ (red). The statistical maps are thresholded with the same convention as Fig. 3.

1) which correlated positively with $G_t$ ($p < 0.001$, uncorrected). These activations were not significant at the whole-brain FWE-corrected at the cluster level ($p$FWE $= 0.69$ for vmPFC and 0.79 for lFPC; Supplementary Fig. 6). In addition, we found that the model estimates of decision difficulty ($|p(C) - 0.5|^{-1}$) did not significantly explain reaction times from the decision onset ($t_{24} = -0.16$, $p = 0.87$), suggesting that participants were more likely to have made a decision when the outcome of the previous interaction was revealed.

**Brain regions tracking updated beliefs about others' choices.** Next, we sought to identify the brain regions computing $\gamma_t$, the belief about others' intention to free-ride—that is the prediction of the likelihood that others would make a free-riding decision at round $t$. Once again, this term was computed when the outcome of the interaction was revealed at trial $t$-1. We found that activity

in the ACCg, $(x, y, z = 0, 20, 31)$ tracked the belief $(\gamma_t)$ about the probability that others would make a free-riding decision (Fig. 4; Supplementary Table 1, $p < 0.05$ FWE corrected at the cluster level). At the same time, activity in bilateral TPJ $(x, y, z = -48, -55, 37)$ and $(51, -55, 40)$ tracked the belief $(1 - \gamma_t)$ of participants about the probability that others would make a contribution at round $t$ (Fig. 4; Supplementary Table 1, $p < 0.05$ FWE corrected at the cluster level; GLM2).

**Neural mechanisms underlying switch between strategies.** One of the strategic decisions in this study is switching one's decision to contribution away from immediate individual utility in favor of the long-term group utility (which indicates collective future expected rewards allocated to not only others but also the player oneself). Because a strategic contribution can induce future contribution of others, switching to contribution is a strategy which potentially leads to greater rewards in the long-term. The other type of strategic decision is switching one's decision from contribution to free-riding to maximize one's immediate reward. To investigate the neural underpinnings of such strategic decision-making, we examined the brain areas showing increased activity for the trials, in which one switches their decision compared with the trials in which one stays with the previous decision[9,23]. We found that strategy switching at round $t$ is predicted by increased activity in the right ventrolateral prefrontal cortex (vlPFC) $(x, y, z = 39, 26, 13)$, and in the ACC $(x, y, z = 0, 17, 31)$ at the time of the outcome of the previous round, $t - 1$ (Fig. 5a; Supplementary Table 1, $p < 0.05$, GLM3, whole-brain corrected with FWE at the cluster level). These areas are therefore likely to be involved in implementing a choice between strategies.

To identify the neural mechanisms underlying the changes in strategies, we examined how the choice probability modulates the interactions between the brain areas involved in switching decisions and the areas involved in encoding the individual utility and the group utility. We hypothesized that brain regions implementing the switch between strategies would show enhanced coupling with those areas encoding subjective utilities. To test for this, we conducted a psychophysiological interaction (PPI) analysis. The physiological variables were the brain signals extracted from the brain areas involved in the switch between strategies (ACC and vlPFC) at the time of feedback. The

psychological variable was the model prediction of the changes in decision value ($\Delta Q$) as a function of to what extent one is more likely to change one's strategy at the trial, $t$ to contribution ($\Delta Q = Q_t - Q_{t-1}$). That is, $\Delta Q$ is positive when one is more likely to change the strategy to contribution, and negative when one is more likely to change the strategy to free-riding, while it is close to zero when one is more likely to stay the previous strategy. The decision value $Q$ was predicted by the social learning model (Eq. (1)).

We found that the vlPFC and the ACC showed increased functional connectivity with the right lFPC, $(x, y, x) = (30, 50, 1)$ ($p < 0.05$, small-volume corrected), the same region encoding the group utility. We also revealed that the vmPFC $(x, y, z) = (9, 50, -8)$, one of the regions encoding the individual utility, showed the opposite pattern of functional connectivity ($p < 0.05$, small-volume corrected). That is, increased functional connectivity was found between the lFPC and the brain areas engaged in switching strategies as a function of the changes in decision value to contribution strategy, while increased functional connectivity was observed between the vmPFC and the brain areas engaged in switching strategies as a function of the changes in decision value to free-riding strategy (Fig. 5b; Supplementary Table 2). Together, these findings suggest that the neural encoding of the group utility and individual utility, as formalized from the social learning model, inform the alteration between strategies in vlPFC and ACC during social interactions (Fig. 5c).

## Discussion

To make a decision within a group, individuals need to estimate both how much utility they can expect when they choose each decision option and the expected utility for their group[2,24]. In the volunteer's dilemma, the decision guided by the strategy to maximize one's immediate expected reward often differs from the decision guided by the other strategy to maximize the long-term collective rewards for the group. Despite the importance of computing utilities of each of the decisions guided by different strategies for collective decisions, little is known about how the human brain encodes each type of utility during repeated social interactions and how they are integrated into a strategic decision. It is therefore a great challenge to understand how the brain maximizes and balances immediate individual utility against long-term group utility when making group decisions for a public

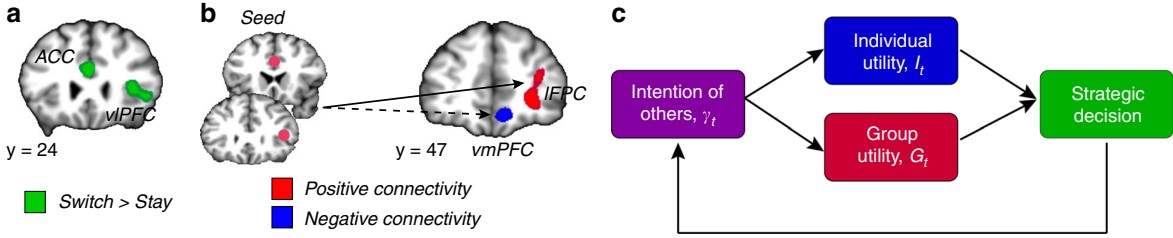

**Fig. 5** Neural mechanisms of strategy selection during the public goods game (PGG). **a** Activity in the right ventrolateral prefrontal cortex (vlPFC) and the anterior cingulate gyrus (ACCg) increased when switching one's decision during the PGG. The statistical maps are thresholded with the same convention as in Fig. 3. **b** Connectivity analyses between the brain regions engaged in switching to the alternative strategy (the vlPFC and the ACC) and the brain areas encoding the individual utility and the group utility. The circles represent seed regions from which physiological signals were extracted, and colored blobs show the psychophysiological interaction effect. The ventromedial prefrontal cortex (vmPFC; blue), encoding the individual utility, shows a negative correlation with the activity in seed regions modulated by the probability to change one's strategy to contribution $\Delta Q$ ($\Delta Q = Q_t - Q_{t-1}$), predicted by the social learning model ($p < 0.05$ small-volume corrected). The right lateral frontopolar cortex (rlFPC; red), which encoded the group utility, shows a positive correlation with the signals in seed regions modulated by $\Delta Q$ ($p < 0.05$ small-volume corrected). **c** Scheme of the social learning model that predicts strategic decisions during PGG. In this model, participants estimate the utility of strategies with the immediate individual utility, $I_t$ and the group utility, $G_t$ used for a long-term strategy based on their belief about the decision of others ($\gamma_t$ and $1 - \gamma_t$). The weighted sum of $I_t$ and $G_t$ is then integrated into a strategic decision, $Q_t$. When the outcome is revealed, the belief about the decision of others ($\gamma_t$) is updated with the social prediction error and the reward prediction error.

good project. Here, we examined the influence of repeated social interactions on strategic decision-making using a threshold PGG, in which the outcome of one's decision depends on the decision of others[5–7].

We showed that the ventromedial prefrontal cortex (vmPFC) encoded this information at the time of feedback of the current interaction, and tracked the relative immediate expected reward of free-riding over that of contribution for the following interactions. It is notable that in the current design of PGG, the relative expected reward of free-riding over contribution varied according to the decision threshold, $k$, and according to one's belief about the decision of others. That is, a volunteer is less likely to contribute to generate the group rewards if there are too many free-riders (more than $N$-$k$ free-riders in $N$ people group), but also if there are too many contributors ($k$ or more than $k$ contributors). These are two cases in which one's contribution does not serve to generate the public goods, but is instead wasted. Therefore, the utility of one's contribution is maximized when participants expected $N$-$k$ people to free-ride. This computational variable of the immediate expected utility depends on one's belief about $N$-$k$ people to free-ride in the following trial ($\Gamma^{N-k}$). In the social learning model, $\Gamma^{N-k}$ is updated based on: the magnitude of rewards allocated to an individual as a result of the current interaction ($R_{t-1}$), one's belief about the decision of another member ($\gamma_t$), and its prediction error. We designed a single model to estimate the brain signal encoding the individual utility and the group utility while these computational variables were competing with other regressors. Using this approach, our results show that the vmPFC activity was specifically encoding individual utility rather than other computational variables. Moreover, we tested the contribution of the individual utility in decision-making by comparing the model predictability of the social learning model to that of the forward-looking model, which did not use the individual utility to predict the decision. Previous studies suggested that the vmPFC reflects the relative value of choice options or the difference in values between options[25,26]. Our current findings extend this function to more complex computations needed in collective decisions, and support the notion that the vmPFC activity reflects computation specific to the individual utility, that is the value of the decision that is about to be executed. Thus, it incorporates not only the expected rewards allocated to an individual but also the influence of one's decision on the rewards allocated to others.

Contrary to free-riding, that benefits individuals with immediate rewards, contribution to the group helps to increase the long-term benefits of the group by fostering their members' cooperativeness. Considering the fact that participants can expect greater long-term utility for the group in a more cooperative group when they have more chance of interactions, we computed the group utility as a function of the number of expected contributors estimated from one's belief about the decision of others ($\gamma_t$) and the number of remaining interactions ($T - t + 1$). Our results demonstrate a role of the lateral frontopolar cortex (lFPC) and the bilateral inferior parietal lobules (iPL) in computing to what extent the contribution decision is worth making when projecting its subsequent effects on expected group utility for the remaining interactions. Importantly, these signals were distinguished from other putative neural correlates of other computational variables which were employed to compute the group utility, such as $\gamma_t$ and $T - t + 1$. Moreover, we tested the contribution of the group utility in decision-making by comparing the model predictability of the social learning model to that of the myopic model which did not use the group utility to predict the decision. The lFPC and iPL both play a critical role in assigning the expected reward to each future trial of social interaction, and use this knowledge to have foresight that guides strategic

decisions[27,28]. Previous studies have suggested that the lFPC computes mounting evidence in favor of an alternative course of action, and that this information is transmitted to the iPL to guide behavioral adaptation[25,29,30]. In those studies, increased activity and functional connectivity in these areas led to switching from the current strategy. Our results show that repeated social interactions can be accounted by model-based reinforcement learning[10,23,31], shedding light on the neurocomputational mechanisms underlying strategic decisions in a group decision-making situation.

When investigating brain areas updating the belief about the decision of another member in the group, we found that the ACCg tracks one's belief about the probability that another member free-rides and that the TPJ tracks the belief that another member will contribute. Such ACCg response may represent an increase in the need to update one's belief about the decision of others when the current belief about the intentions of another player to make a contribution is more volatile[12,32]. Many neuroimaging studies in humans have found ACC activation during social interactions[32,33]. Sub-regions of ACC have distinct cytoarchitecture and connectivity, suggesting that they play different roles in social cognition[33–37]. Using single-unit recordings, ACCg neurons have been shown to respond to rewards allocated to others[38,39]. In functional neuroimaging studies, the ACCg also tracks the decision of others[40], and its activity increases with the volatility or unpredictability of another's intention[13]. The TPJ has been shown to signal "other-oriented" information and has strong connections to the ACCg[34,37,41]. During social interactions, the TPJ has been reported to be engaged when inferring the mental states of others[10,38,42]. Our results support the view that the ACCg and TPJ compute one's belief about the intention of others. More importantly, they demonstrate that these brain areas monitor different types of strategy. Such increased TPJ activity might be useful when interacting within a more cooperative group, thereby promoting greater willingness to make altruistic decisions[42].

In this study, participants who adopted a mixed strategy might flexibly switch their strategies between free-riding and contribution according to the changes in their expected utility. Considering strategic decision-making as the flexible shifts between different strategies, we observed that the activity from the vlPFC and the ACC increased when switching between strategies, suggesting that these brain regions signal the needs to change the current strategy during collective decisions. It is important to note that here, we defined a change in strategic-decision as a change in model-predicted decision value, because it is not possible to dissociate a change in strategy from a change in behavioral response. This interpretation relates to the results of a previous study, suggesting that the vlPFC is engaged in controlling model-based and model-free decision strategies[23]. Also, the functional connectivity of the ACC to other brain areas tracking the history of others' decisions and one's own preference has been shown to guide collective decisions during consensus decision for a group[43]. Moreover, the activity of the ACC correlates with individual differences in the degree to which an individual prefers one strategy over the other during competitive decision-making[44]. The increased strength of the relationship between ACC and lFPC may relate to a central role of the dACC–dlPFC interactions proposed in relation to a key decision variable called prospective value, as opposed to an immediate myopic value[45,46]. In such framework, when making sequential decisions, the overall value of the environment can be decomposed into a myopic component, corresponding to the average benefits that might immediately follow a decision, and prospective value, corresponding to future benefits that might accrue over the longer-term by taking a particular choice now[45]. In light of this previous study, our results suggest that the strength of the dACC–lFPC connectivity

increasing with group utility may reflect choice strategy related to prospective valuation.

To conclude, we have decomposed the neural mechanisms required to enable efficient allocation of one's contribution in a group decision-making situation known as the volunteer's dilemma. When the outcome of one's decision is determined by the decision of others, the prefrontal cortex computes individual utility and group utility, respectively, in the vmPFC and in the lFPC. In doing so, the value of each state in future interactions is updated according to the changes in one's belief about the decision of others. Moreover, the ACCg and TPJ tracked the beliefs about the decision of another member in the group. Finally, functional connectivity between areas (ACC and vlPFC) guiding switch to different strategies and brain areas computing individual (vmPFC) and group (lFPC) utilities were modulated by the probability to change one's current strategy to contribution, revealing the neural mechanisms underlying the adoption of a mixed strategy. Together, these findings provide a mechanistic framework for the computations underlying strategic collective decisions.

## Methods

**Participants**. We recruited 30 right-handed university students. They participated in the threshold PGG and made decisions in a 3T Philips Achieva MRI scanner. Because of excessive head movement and lower social contextualization score in the post-scanning questionnaire (Supplementary Methods, lower than three times of the standard deviation), data of five participants were not included for analysis. The final sample consisted in 25 participants (mean age 22.48 years old ± 0.33, 13 women). Based on self-reported questionnaires, none of them reported a history of neurological or psychiatric disorders. This study was approved by the Institutional Review Board of the local ethics committee (IRB n°A13-37030), and all participants gave their informed written consent.

**Threshold PGG**. In the PGG used in this study, a participant was led to believe that he made decisions within a group of five members ($N = 5$). The participants were told that they would play with 19 other participants located in another room, so that 20 participants in total would play the PGG in four different groups of five subjects simultaneously, randomly arranged by a computer at the beginning of every PGG. Hence, participants lying in the scanner knew that they were interacting with a different combination of group members at the beginning of every PGG.

A PGG was composed of 15 rounds of interactions with the same partners ($T = 15$). (1) At the beginning of each PGG, participants were informed that they were grouped with new members. (2) At the beginning of each round, 1 MU was endowed to every individual in the group. (3) Participants were asked to make a binary decision simultaneously with others. They decided between contribution ($C$) with 1 MU cost and keep to make a free-riding ($F$) to maximize their own utility (Fig. 1a). Participants had to decide within 3 s. Otherwise, a warning message appeared, and the trial was repeated. After all members of a group made decisions in a given trial, a feedback was given to all members of the group. (4) Participants got feedbacks about the number of contributors (as well as number of free-riders) and whether they succeed to generate the group rewards. According to the decision of group members, public goods were produced as group rewards ($R = 2$ MU) only if at least $k$ individuals contributed their resources. The $k$ was clearly indicated in the center of the screen to all group members before making their decisions, and was kept constant during a single PGG ($k = 2$ or $k = 4$). During experiments, any kind of communication among members was not allowed.

Before playing the 12 PGGs, participants played two additional non-feedback PGGs (control block), in which we estimated the initial tendency to make contribution before experiencing social interactions. The distribution of the initial contribution rate is shown in Supplementary Fig. 2. Anatomical scans were acquired during the control blocks. While playing 12 blocks of PGG, participants were provided with feedback. The behavioral and fMRI analyses were made with the data acquired during these 12 test blocks. At the time of feedback, the decision of others was represented by the position (left or right for keep or contribute) of a generic white icon of a person, so participants could not track the decision of specific other individuals (i.e., anonymous interaction). In addition, the position of a yellow icon of a person (left or right for keep or contribute) represented the decision of the participant. The contribution decision was shown on the right or left side of the screen, and such position was counterbalanced across participants. Moreover, the current round ($t$) and the remaining numbers of interactions ($T - t + 1$) were graphically shown to participants to inform the progress of the PGG. The accumulated payoffs were only shown at the end of each PGG.

The payoff was calculated as follows. 1 MU is given per every trial as endowment ($E$). The contribution had a fixed cost ($c = 1$ MU). Therefore, when public goods are produced ($R = 2$ MU), the expected payoff is: $E - c + R = 2$ MU for the contributor and $E + R = 3$ MU for the free-rider. When public goods are not produced (the group reward is 0 MU), the expected payoff is $E - c = 0$ MU for the contributor and $E = 1$ MU for the free-rider. Figure 1b shows the expected payoff matrix in a round of the PGG. Critically, in PGG, the payoff depends not only on one's own decision but also the decision of others. Accordingly, Supplementary Fig. 1a illustrated the four examples of possible results paying out different amounts to a participant from 0 MU to 3 MU. These rules were given to all participants before the experiment. Importantly, we also informed participants that they would get a final monetary reward, which could be as much as the result of one PGG (randomly selected by a computer to make each PGG independent).

**Decisions of other members of the group**. To control the underlying motivations of other individuals across participants while creating plausible behavior in social interactions, unbeknownst to the participants, decisions of other members of the group were determined by a computer algorithm. First, the probability that a computer agent contributed in the first round was determined by the proportion of contribution decisions made by each of the participants during the control condition of PGG, in which no feedback was given. That is, the computer agent contributed as much as the participant did. Second, the decision of the computer agent was determined by their and others' decision ($C_{t-1}^i$ and $C_{t-1}^{N-1}$, respectively) in the previous round, the decision threshold ($k$), and the number of remaining interactions ($T - t + 1$, where $t$ is the current trial) with a weight ($\beta$), which determined to what extent the agents change their decision according to the decision of others, or stay with their previous decision. Third, $\beta$ was determined by the value which gave the maximum likelihood while predicting the actual decisions made during real human interactions in a previous study[22]. As a result, the computer agent tended to stay with their previous decisions, he/she was more likely to contribute in a more cooperative group, and he/she was more likely to free-ride when failing to generate the public goods in a less cooperative group. At last, with the post-scanning questionnaire, we ensured that we analyzed the data acquired from the participants who had believed that they had interacted with real human participants simultaneously (see the Supplementary Methods for social contextualization score). The details of the computer algorithm generating the decision of others are described in the Supplementary Methods.

**Model-free behavioral analysis**. We predicted the probability of contribution, $p(C_t)$ using a multiple logistic regression with the trial history of the number of free-riders among others ($nF$), participant's previous decision, success or failure of producing public goods and the interaction of decision and success of producing public goods. We entered information of multiple previous trials as regressors (from $t - 1$ to $t - 3$), which included the information acquired from the 1st to the 14th trials to explain the decision at the trial $t$ (including the 4th to the 15th trials, because decision at trial $t$ is predicted by information acquired from $t - 1$; $t - 2$; $t - 3$). To treat the subjects as random effects, we performed the one-sample $t$ test with the coefficients assigned to each regressors across participants.

**Social learning model**. The goal of the computational model is to predict the contribution decision at trial $t$ while a participant plays a threshold PGG with repetition ($t = [1:15]$). In order to determine key computational variables involved in decision-making when confronting to a certain level of volunteer's dilemma, we built a family of computational models and fitted those models to the participants' actual choice behavior. The decision value at round $t$, called $Q_t$, was estimated by the weighted sum of individual utility ($I_t$) and group utility ($G_t$). We assumed that participants decided to contribute to maximize their subjective utility of expected outcome. Therefore, the decision value, $Q_t$ governs the participant's probability of contribution, $p(C_t)$ as follow:

$$p(C_t) = \text{logit}(Q_t)$$
$$Q_t = \omega I_t + (1 - \omega)G_t \tag{1}$$

**Immediate expected utility for an individual ($I_t$)**. $I_t$ is computed as previously proposed by a model[4,47] that provided the mixed-strategy equilibria for a group, in which individuals interacted only once with volunteer's dilemma during one-shot PGG. $I_t$ is defined as a relative utility on the current trial between two decision options. We expect that participants tend to contribute more when expecting the utility of contribution, $I(C_t)$ is at least the same as the expected utility of free-riding, $I(F_t)$.

$$I_t = I(C_t) - I(F_t) \tag{2}$$

When $R_C$ indicates the outcome of the group when participants contribute and $R_F$ indicates the outcome of the group when participants free-ride, both $R_C$ and $R_F$ can have the same value of 2 MU or 0 MU ($R_C = R_F$) or be different from each

other ($R_C = 2$ MU; $R_F = 0$ MU; $R_C \neq R_F$) according to the decision of others. Furthermore, the expected reward of contribution ($\mathbb{E}(C_t)$) and that of free-riding ($\mathbb{E}(F_t)$) allocated to the participants (called "one") are determined by their belief about $\Gamma^i$, which was defined as the probability that the number of free-riders among $N - 1$ others will be $i$ at round $t$ ($i$ is the number of free-riders among others; $0 \leq i \leq N - 1$).

$$\mathbb{E}(C_t)_{\text{One}} = \lambda + \sum_{i=0}^{N-1} \Gamma^i (R_C)$$
$$\mathbb{E}(F_t)_{\text{One}} = \sum_{i=0}^{N-1} \Gamma^i (R_F)$$

(3)

where $\lambda$ indicates the subjective value of the contribution cost ($\lambda < 0$). That is, participants who estimate the more expensive contribution cost are more reluctant to contribute.

In addition to the payout given to herself, a participant might also consider the payout given to other fellows in the group as her altruistic reward and integrates it into $I_t$. When $i$ members free-ride among $N - 1$ others, the number of contributors among others equals $N - 1 - i$. Given that every member gets the same amount of group reward ($GR = R_C$ or $GR = R_F$), the expected payoff of the contributor is $E + GR - c$, and the expected payoff of the free-rider is $E + GR$ (endowment ($E$) = 1 MU; contribution cost ($c$) = 1 MU). Therefore, one's expected rewards given to other members when she makes a contribution, $\mathbb{E}(C_t)_{\text{Others}}$ and those when she makes a free-riding decision, $\mathbb{E}(F_t)_{\text{Others}}$ are dependent on one's decision: contribution ($GR = R_C$); free-riding ($GR = R_F$) and her belief about the decision of others, $\Gamma^i$.

$$\mathbb{E}(C_t)_{\text{Others}} = \sum_{i=0}^{N-1} \Gamma^i \{(N - 1 - i)(E + R_C - c) + i(E + R_C)\}$$
$$\mathbb{E}(F_t)_{\text{Others}} = \sum_{i=0}^{N-1} \Gamma^i \{(N - 1 - i)(E + R_F - c) + i(E + R_F)\}$$

(4)

The expected utility given to the other members comes to the altruistic reward, which is also added to one's expected utility while considering the different tendency to make an other-regarding decision. This tendency is captured with the parameter, $\pi$ as follows: $\pi < 0$ for who decides to have more rewards than others; $\pi = 0$ for who is indifferent to the payout of others; $\pi > 0$ for who would make an altruistic decision (Supplementary Fig. 3). Therefore, the relative utility between two possible strategies is computed as below:

$$I(C_t) - I(F_t) = (\mathbb{E}(C_t)_{\text{One}} - \mathbb{E}(F_t)_{\text{One}}) + \pi(\mathbb{E}(C_t)_{\text{Others}} - \mathbb{E}(F_t)_{\text{Others}})$$
$$= \lambda + \sum_{i=0}^{N-1} \Gamma^i (R_C - R_F) + \pi\left\{\sum_{i=0}^{N-1} \Gamma^i \{(N - 1 - i)(R_C - R_F) + i(R_C - R_F)\}\right\}$$

(5)

During the PGG, the outcome of one's decision is dependent on the decision of others given one's decision. For the PGG where the decision threshold is $k$, if $i$ people decide to free-ride among $N - 1$ others, the group outcome—when participants free-ride, $R_F$ and when participants contribute, $R_C$ — becomes 2 MU or 0 MU as follows.

$$R_F = \begin{cases} R, & \text{if } (N - 1) - i \geq k \\ 0, & \text{if } (N - 1) - i < k \end{cases}$$
$$R_C = \begin{cases} R, & \text{if } (N - 1) - i \geq k - 1 \\ 0, & \text{if } (N - 1) - i < k - 1 \end{cases}$$
$$R_C - R_F = R, \text{ when } i = N - k (R_C \neq R_F)$$
$$R_C - R_F = 0, \text{ otherwise } (R_c = R_F)$$

(6)

That is, the outcome would be the same ($R_c = R_F$; $R_C - R_F = 0$) in most of cases. However, when the number of free-riders among others, $i$ equals $N - k$, $R_c$ is $R$ (2 MU) and $R_F$ is 0 MU ($R_c \neq R_F$; $R_C - R_F = R$). Therefore, the relative utility between two possible strategies is simplified as:

$$I_t = \lambda + \Gamma^{N-k} R + \pi \Gamma^{N-k} R (N - 1)$$

(7)

**Long-term expected utility for the group ($G_t$).** The computational model of $G_t$ was adopted from a previous theoretical study that predicted the contribution decisions in the PGG, in which one interacts with the same partners with finite repetition[17]. Note that this previous version of the PGG was not confronted by volunteer's dilemma. For the one-shot interaction, participants can expect higher payoffs when free-riding regardless of the decision of others. The free-riding decision that causes a greater immediate reward, however, decreases the chance of the group to generate public goods, which turns other members to be less likely to contribute. The more cooperative the group, the greater benefits an individual can expect, which is true in many cases of social interactions. Taking the effects of one's current decision in the generation of long-term rewards into account, therefore, is critical for successful strategic decision-making. We expect that participants are more likely to expect public goods after successful cooperation of that group. Taking this into account, we computed $G_t$ as representing the cumulative expected rewards of the group for the remaining interactions. The pending reward from future interactions is estimated based on one's belief about the extent that the group would generate public goods. Therefore, if $\Gamma^i$ indicates one's belief about the

probability that $i$ people free-ride among $N - 1$ people, $G_t$ follows the function:

$$G_t = \sum_{j=t}^{T} RK^{T-j} \sum_{i=0}^{N-k} \Gamma_t^i$$
$$= \frac{1 - K^{T-t+1}}{1 - K} R \sum_{i=0}^{N-k} \Gamma_t^i$$

(8)

where $K$ is the ratio between the stated minimum number of contributors required for generating public goods and the size of the group ($K = k/N$), and $R$ is the magnitude of the benefits from public goods ($R = 2$ MU). Importantly, the function has a weight that decays $G_t$ with the remaining trials after the interaction at the trial $t$ ($T - t + 1$). This decay function allows us to capture that $G_t$ is high at the beginning of the PGG when one's decision influences relatively many future interactions, but $G_t$ is discounted as PGG progresses.

**The beliefs about the decisions of others.** Two forms of utility expected by choosing a strategy—$I_t$ and $G_t$—were dependent on the belief about the probability that one of the other player free-rides at round $t$, $\gamma_t$. Given that the decisions of others are revealed anonymously in this game, participants are not able to track the probability of each of others free-riding decision at round $t$. Instead, they may compute that all have the same level of probability to free-ride. The probability that $N - k$ individuals among $N - 1$ others would free-ride follows the probability mass function of a binomial distribution. Therefore, the probability that $N - k$ individuals would free-ride is computed as follows given that $\gamma_t$ is the probability of free-riding of one of the other members at round, $t$:

$$\Gamma_t^{N-k} = \binom{N-1}{N-k} \gamma_t^{N-k} (1 - \gamma_t)^{k-1}$$

(9)

We assumed that participants update their belief $\gamma_t$ by learning from the social prediction error ($PE_S$) using a reinforcement learning algorithm. The $PE_S$ is the difference between the prediction and the revealed probability of others to free-ride at round $t - 1$. Specifically, $PE_S$ was computed as the difference between the proportion of the free-riders of others members at round $t - 1$, that is ($F_{t-1}/N - 1$) and one's prediction of it, $\gamma_{t-1}$. The $PE_S$ is updated with the learning rate $\alpha$, and the learning rate increases when there is a reward prediction error ($PE_R$). Therefore, the learning rate to update the belief about the free-riding of others is $\alpha + \theta \times PE_R$. $PE_R$ is the absolute value of the difference between the actual outcome of the interaction at round $t - 1$, $R_{t-1}$ and the expected reward at round $t - 1$. The expected group reward is computed as the magnitude of the public goods ($R = 2$ MU) weighted by the probability that the group generate public goods.

$$\gamma_t = \gamma_{t-1} + f(\alpha + \theta PE_R)PE_S$$
$$\text{where } PE_S = (F_{t-1}/N - 1) - \gamma_{t-1}$$
$$\text{and } PE_R = \left| R \sum_{i=0}^{N-k} \Gamma_{t-1}^i - R_{t-1} \right|$$

(10)

where the learning rate is a function of $PE_R$ and $f$ is a logistic function. We set the initial belief of each participant about the probability of one of the others to free-ride ($\gamma_1^k$) as the proportion of free-riding decisions of trials in non-feedback PGG (control conditions) based on the assumption that participants expect the others will decide as they would. The distributions of $\gamma_1^k$ are shown in the Supplementary Information (Supplementary Fig. 4). In addition, to test whether participants vary their learning rate in proportion to the reward prediction error, we tested whether the weight, $\theta$ is greater than zero across participants.

The contribution decision can also be influenced by one' initial belief about decisions of others before experiencing any interaction. Note that other modeling approaches of these behavioral data are possible. In particular, a recent study used Partially Observable Model Decision Processes inferred the latent initial belief of each participant[48]. Using the individual differences in this initial belief as the prior, this study assumes that every participant updates the belief about decision of others while observing interactions in Bayesian manners. In this study, instead, we performed an additional control experiment to include the effect of the initial belief of participants in the model. We inputted one's mean contribution rate as the initial belief of each participant while playing the PGG in which they did not get any feedback (see the control PGG in the Methods section). Our models focused more on how the individual participant uses the previous interactions to compute the utility of different strategies. With this model, here we address the question about how the human brains make a strategic decision while interacting with the same partners.

**Testing higher-order beliefs on other people's beliefs.** Previous findings have suggested that participants might adopt iterative reasoning, forming higher-order beliefs on other people's beliefs when predicting others' behavior in strategic interactions[11,18,19]. To explore if higher-order beliefs (e.g., belief of other people on the belief of other people) explain participants' behavior better, we compared the likelihoods of the social learning model while modulating the level of iterative reasoning.

If participants use the 2nd order belief model, then, they think that another player will use the 1st order belief model for free-riding with probability $\gamma_t^{1st} = \Gamma_t^{N-k}$, because these players who used the 1st belief model think that others

will make free-riding with probability, $\gamma_t$. Therefore, the decision of players who use the 2nd order belief model is predicted based on their belief about the decision of another as $\Gamma_t^{N-k}$ instead of $\gamma_t$.

Likewise, if the participants use the 3rd order belief model, they think that another player will free-ride with probability, $\gamma_t^{2nd}$ given the assumption that another player thinks that others will free-ride with probability $\gamma_t^{1st} = \Gamma_t^{N-k}$. To examine whether higher-order beliefs explain participants' behavior better, we compared the predictability of the models assuming different levels of iterative reasoning. Specifically, we replaced $\gamma_t$ in the social learning model (Eq. (11)) with $\gamma_t^{1st}$ for the 2nd order belief model and with $\gamma_t^{2nd}$ for the 3rd order belief model where $\gamma_t^{1st}$ and $\gamma_t^{2nd}$ are computed as follows:

$$\gamma_t^{1st} = \Gamma_t^{N-k} = \binom{N-1}{N-k}\gamma_t^{N-k}\left(1-\gamma_t\right)^{k-1}$$
$$\gamma_t^{2nd} = \binom{N-1}{N-k}\Gamma_t^{N-k^{N-k}}\left(1-\Gamma_t^{N-k}\right)^{k-1} \quad (11)$$

**Parameter estimates.** We estimated the probability of each participant to contribute during round $t$ during the PGG ($p(C_t|\alpha,\theta,\omega,\pi,\lambda)$), as well as the parameters that maximize the likelihood function by fitting this model to the participants' actual decisions ($C_t$). The likelihood function is as follows:

$$\ln\hat{L} = \sum C_t \times \ln p(C_t|\alpha,\theta,\omega,\pi,\lambda) + (1-C_t)\times\ln(1-p(C_t|\alpha,\theta,\omega,\pi,\lambda)) \quad (12)$$

**Alternative models.** We compared the goodness of fit of this model with alternative models. The first alternative model is the "myopic model" (individual utility model), which can be distinguished from the "social learning" model because it assumed that participants only take their $I_t$ into account. Based on this model, the contribution decision is dependent on the immediate expected reward, and the decision value $U_t$ is determined by five free parameters as in the "social learning" model:

$$p(C_t) = \text{logit}(U_t) = \text{logit}(\omega I_t) \quad (13)$$

The second alternative model is the "group utility model" which can be distinguished from the "social learning" model because it only takes Gt into account. In doing so, this model assumed that participants are more likely to contribute in the group where they can expect high mutual contribution. In this model, the probability of making a contribution depends on the decision value $V_t$, which includes four free parameters, including X as the weight assigned to the group utility and $\zeta$ as the initial bias (error term).

$$p(C_t) = \text{logit}(V_t) = \text{logit}(\zeta + \chi G_t) \quad (14)$$

We also tested an "inequity aversion model", in which participants continue to contribute with a certain probability, $p(C_1)$ until their group generates public goods ($R_t$), but their contribution rate decays with probability, $p(W_t)$ after they contribute ($C_i$) more than the average contribution of others ($\bar{C}_i$). The probability, $p(W_t)$ is defined by three parameters that take account the individual difference in the sensitivity to the reward, $\varepsilon$, the level of inequity, $\delta$, and the inverse temperature parameter $\kappa$. To what extent an individual is willing to contribute to the group was measured by the rate of contribution decisions during the no-feedback PGG ($p(C_1) = 1 - p(F_1)$).

$$p(C_t) = p(C_1) \times p(W_t)$$
$$\text{where } p(W_t) = \text{logit}\left(\kappa\sum_{i=1}^{t-1}\varepsilon R_i - \delta(C_i - \bar{C}_i)\right) \quad (15)$$

where the inverse temperature parameter $\kappa$ sets the level of noise in the decision process, with large $\kappa$ corresponding to a more deterministic decision based on decision value with a lower decision noise ($0 < \kappa < \infty$).

**Synthetic data generation for model validation.** We generated a computer agent whose decision is determined by the social learning model with free parameters guiding the decision estimated from the actual decisions made by participants while playing other 11 PGGs in the scanner (total 12 PGGs). Therefore, we predicted how the agent would decide for a PGG (test set) by learning the decisions of 11 training sets. This procedure was repeated 12 times per subject to generate whole sets of synthetic decision for 12 PGGs for the same number of participants. With these synthetic datasets, we first tested whether the data generated by the social learning model capture the model-free characteristics of strategic decision-making. Second, we tested the changes in quality of model fits by removing each of four free parameters ($\alpha$, $\theta$, $\lambda$, and $\pi$) to evaluate its contribution qualitatively. With these methods, we tested the validity of the social learning model.

**Model comparisons.** The goodness of fit was assessed by the penalized likelihood using Bayesian information criterion (BIC) that considered the number of free parameters in each model:

$$\text{BIC} = -2\ln\hat{L} + \mu\ln(n) \quad (16)$$

where $\mu$ is the number of free parameters in each model and $n$ is the number of observations of the event. We compared and reported the sum of BIC of every trial across participants. In addition to BIC, we also computed the posterior expectation of the likelihood of each model using Bayesian model selection to compare the goodness of model fits[49].

**fMRI data acquisition.** Functional images were acquired with a Discovery MR750 MRI scanner (General Electric, Milwaukee, WI, USA) operating at 3 Tesla in the University Hospital of Parma, Italy. The imaging parameters were as follows: repetition time (TR), 2500 ms; echo time (TE), 30 ms; acceleration factor 2, bandwidth 3906 Hz/PIXEL matrix $96 \times 96$, field of view (FOV), $205 \times 205$ mm$^2$; 41 contiguous slices were acquired in interleaved order, slice thickness, 2.8 mm + 0.7 mm gap. The imaging parameters for the 3D IR-prepared FSPGR T1-weighted anatomical scan were as follows: TR, 8500 ms; TE, 3.2 ms; FOV, $256 \times 256$ mm$^2$; matrix $256 \times 256$; slice thickness, 1 mm; total slices, 156, bandwidth 244 Hz/PIXEL.

The stimuli were presented with a head-mounted VisualStim system goggles (Resonance Technology, San Diego, CA, USA) with a screen resolution of $800 \times 600$ pixels and surrounded by a black background. A fiber optic Response Box device was used to measure the responses (Resonance Technology, San Diego, CA, USA). The participants were asked to use their index and the middle fingers to answer by pressing two buttons. The stimuli presentation and the responses collection were done using the Presentation software (Neurobehavioral Systems, CA, USA).

**fMRI data analysis.** Image preprocessing was performed using SPM8 (Wellcome Trust Centre for Neuroimaging, UCL, UK). Time-series images were registered in three-dimensional space to minimize any effects that could result from the motion of the participants' heads. Functional scans were realigned to the last volume, corrected for slice timing, co-registered with structural maps, spatially normalized into the standard Montreal Neurological Institute (MNI) atlas space, and then spatially smoothed with an 8- mm isotropic full-width at half-maximum (FWHM) Gaussian kernel using standard procedures.

We ran general linear model (GLM) analyses to identify which brain regions encode the following computational variables: estimates of the individual utility (I), the group utility (G), and one's belief about the decision of others ($\gamma$). These computations are serving to make decision at trial, $t$. In addition, we allow these regressors to compete with other regressors which are serving to process the outcome of the previous interaction, $t − 1$: the weighted prediction errors (wPE, the update term in Eq. (10)) and the reward allocated to the participant (R). To control for the number of remaining trials, we also inputted the trial number, $t$ as an additional regressor. First, we examined the brain signals encoding I and G (GLM1), and examined the brain regions computing $\gamma$ (GLM2) while controlling for the covariation between regressors. To deal with the multicollinearity issue (Supplementary Fig. 5a), we inputted the regressors of interest (I and G for GLM1 and $\gamma$ for GLM2), and we inputted the other regressors after regressing out their shared variance with the regressors of interest by performing a partial correlation.

Specifically, for GLM1, which serves to identify the brain regions encoding individual utility (I) and group utility (G), we included the parametric regressors of utilities (I and G), as well as wPE$^{IG}$, R$^{IG}$, t$^{IG}$ and $\gamma^{IG}$. The parametric regressor, wPE$^{IG}$ was computed as wPE $− I\beta_{wPE,I} − G\beta_{wPE,G}$ where $\beta_{wPE,I}$ and $\beta_{wPE,G}$ indicate to what extent wPE was explained by variances of I and G, which was also applied for computing R$^{IG}$, t$^{IG}$, and $\gamma^{IG}$. This partial correlation allowed us to control for the confounding variable of other regressors (wpE$^{IG}$, R$^{IG}$, t$^{IG}$, and $\gamma^{IG}$) by preferentially assigning covariance to regressors of interests without any transformation (I and G). Importantly, wPE$^{IG}$, R$^{IG}$, t$^{IG}$, and $\gamma^{IG}$ are still highly correlated with wPE, R, t, and $\gamma$, respectively (see orange colored area in Supplementary Fig. 5b), while they do not correlate with I and G anymore (see purple colored area in Supplementary Fig. 5b). This partial correlation method allows us to prioritize multiple regressors of interests equally (I and G in GLM1) and identify brain activity specifically correlating with each of computational variables while controlling their covariance with other regressors (wPE, R, t, and $\gamma$ in GLM1). The advantage of this method over the classical orthogonalization method is that the latter would only allow us to keep one regressor's value and to change the values of sub-rank regressors sequentially according to their priority.

For GLM2, which serves to identify brain activity specifically encoding $\gamma$, we included $\gamma$, I$^\gamma$, G$^\gamma$, wPE$^\gamma$, t$^\gamma$, and R$^\gamma$. As described above, I$^\gamma$ was computed as I $− I\beta_{I,\gamma}$ where $\beta_{I,\gamma}$ indicates to what extent I was explained by $\gamma$. In this way, again, I$^\gamma$, G$^\gamma$, wPE$^\gamma$, t$^\gamma$, and R$^\gamma$ are still highly correlated with I, G, wPE, t, and R, respectively (see orange colored area in Supplementary Fig. 5c), while they do not correlate with $\gamma$ anymore (see purple colored area in Supplementary Fig. 5c). We thus controlled the confounding variables of other regressors (I$^\gamma$, G$^\gamma$, wPE$^\gamma$, t$^\gamma$, and R$^\gamma$) by prioritizing the effects of regressor of interest ($\gamma$).

The participant-specific design matrices contained the boxcar functions of outcome presentation (from its onset with 4 s duration) from the 1st to the 14th rounds to examine brain activity involved in decision-making for the 2nd to the 15th trials of PGG. Additional regressors of non-interests were as follows: a stick function for the button press onsets and the decision onsets; a boxcar function of error message presentation—trials in which subjects or their team member did not respond— which was modeled as separate regressors (3 s). In addition, the motion

parameters produced by head movement were also entered as additional regressors of no interest to account for motion-related artifacts.

Last, we examined the brain regions specifically engaged when switching one's decision strategy from one trial to another, compared with staying with one's previous strategy (Switch > Stay). For the third GLM (GLM3), we thus compared the brain responses of the outcome at round $t-1$ (including the 1st to the 14th round of the PGG) to model the decision at the round $t$. The outcome phases of the round $t-1$ of the PGG were split into the "switch" and the "stay" trials according to the interaction between the decision at round $t-1$ and the decision at the following round $t$. The switch and stay trials were determined based on the decisions made by each participant, and were not based on the model-based prediction. GLM3 also included the same regressors of no interest defined in GLM1, which includes button press, instructions, decision onsets, missing trials, and motion regressors.

All of the regressors were convolved with a canonical hemodynamic response function. In the model specification procedure, serial orthogonalization of parametric modulators was turned off. Contrast images related to modulation of brain activity at the time of feedback by the model-based predictions of the changes in the computational variables (GLM1 and GLM2) and the contrast images related to switching the current strategy (Switch > Stay; GLM3) were calculated and entered in a second level analysis (one-sample $t$-test).

We reported brain areas showing significant activity at the threshold of $p < 0.05$, whole-brain family-wise error (FWE) corrected for multiple comparisons at the cluster level.

**Psychophysiological interaction (PPI) analysis**. We define the psychological factor of strategic decisions as the trials requiring the switch in strategy. Based on this definition, there are two types of strategic decisions. One of the strategic decisions is to switch one's free-riding (Fr) decision to contribution (Co) in favor of the group utility (Fr → Co). Since a strategic contribution can induce future contribution of others, switching to contribution is a strategy which potentially leads to greater rewards in the long-term. The other type of strategic decision is to switch one's contribution decision to free-riding to maximize one's immediate reward (Co→Fr). With this definition of strategic decisions, we were able to examine the functional connectivity specific to the decision of switching to free-riding and the decision of switching to contribution compared with the decision of staying with the previous decision.

Using a PPI analysis, we focused on the time of feedback and examined the changes in functional connectivity between the brain regions involved in the switch in strategies and the brain regions encoding the computational variables (the individual utility, $I_t$ and the group utility, $G_t$). For the PPI, we defined the seed regions of interest (ROIs) as a 8-mm radius spheres centered on the coordinates extracted from the peak voxel of the right ventrolateral prefrontal cortex (vlPFC), $(x, y, z = 39, 26, 13)$ in MNI coordinates and the anterior cingulate cortex (ACC), $(x, y, z = 0, 17, 31)$ predicting the changes in strategies (GLM3). The physiological variable is therefore the time series extracted from a priori ROIs at the time of feedback phase of $t-1$ trial (the 1st to the 14th rounds) of the PGG. In addition, we defined the psychological factors as the model prediction of the probability to switch one's strategy to contribution. Specifically, it was computed as the difference in the decision value ($\Delta Q = Q_t - Q_{t-1}$) (Eq. (1)). The decision value, $Q$ is predicted from the social learning model. That is, participants tend to switch to contribution strategy when $\Delta Q > 0$, while they tend to switch to free-riding strategy when $\Delta Q < 0$, and they tend to stay to their current strategy when $\Delta Q \approx 0$. The GLM for the PPI analysis therefore contained the following regressors: (1) physiological factors, BOLD signals from the ROIs, (2) psychological factors, and (3) PPI factors, interaction terms of the psychological and physiological factors, as well as the same regressors of no interests that we used for GLM1.

The statistical significance of the functional connectivity in the vmPFC and that in the right lFPC were tested within anatomically defined independent ROIs. The ROIs were defined by a previous study that anatomically parcellated the prefrontal cortex according to the resting state connectivity[50]. Specifically, we, respectively, used two parcellations annotated as area "11 m" and the "frontopolar cortex lateral" (FPl).

**Reporting summary**. Further information on research design is available in the Nature Research Reporting Summary linked to this article.

## Data availability
Behavioral data are available via the Open Science Framework with the identifier https://doi.org/10.17605/OSF.IO/RDVSZ. Unthresholded group-level statistical maps are available on NeuroVault (https://neurovault.org/collections/QXJNIAXH). A reporting summary for this Article is available as a Supplementary Information file.

## Code availability
Code supporting our main analyses are available via the Open Science Framework with the identifier https://doi.org/10.17605/OSF.IO/RDVSZ.

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

## Acknowledgements
This work was funded by the NSF-ANR Collaborative Research in Computational Neuroscience 'CRCNS SOCIAL_POMDP' n°16-NEUC to JCD. This research was performed within the framework of the Laboratory of Excellence "LABEX ANR-11-LABEX-0042" of Université de Lyon, attributed to JCD, within the program "Investissements d'Avenir" (ANR-11-IDEX-0007) operated by the French National Research Agency (ANR). J.C.D. has also benefited from the financial support of IDEXLYON from Université de Lyon (project INDEPTH) within the "Programme Investissements d'Avenir" (ANR-16-IDEX-0005). M.S. was supported by a post-doctoral fellowship from the Fyssen Foundation.

## Author contributions
S.A.P. and J.-C.D. designed the experiment; M.S. acquired the data. S.A.P. analyzed the data; S.A.P., M.S., E.B. and J.-C.D. wrote the paper. All authors contributed to the interpretation of the results and edited the paper.

## Competing interests
The authors declare no competing interests.
