## [Peer Review File · Nature Communications]

Reviewers' Comments:

Reviewer #1:

Remarks to the Author:

In this paper the authors investigate social decision-making using a iterated Public Goods Game (PPG), in which players had to give up their initial endowment toward a public good that would benefit all 5 players. However, the public good (or common reward) would only be realized if at least 2 or 4 players would contribute, whereas the other players could free-ride, thus keeping their endowment and, if the public good was realized, also received the common reward. In the modeling the authors claim that a social learning model that balances individual and group utility provides the best fit to the data. In the model-based fMRI analysis, they found several brain region (ACC, dmPFC, IPFC, vmPFC, rTPJ) - known from other studies on social decision-making - to be involved in representing and integrating key model-based signals.

The topic of this study (strategic social decision-making) is timely and relevant to broad readership of Nature Communication. The experimental design of the study is good and capable of producing interesting data. However, there are many inaccuracies and even errors in the modeling and fMRI analysis (detailed below) some of which are likely going to have an impact on the results. This makes this paper unsuitable for a top-level journal like Nature Communications.

Behavior

Figure 1D show basically the same thing in 3 different panels, namely that participants contribute less in $k=2$ trials and that their probability of contribution declines across the PGG. I like the trial-by-trial display, but it would be more informative to combine these data with the success of the subjects (just as in the companion paper using POMDPs (Khalvati et al., <https://www.biorxiv.org/content/early/2018/09/17/419515>). This could reveal an interesting pattern in the model-free behavior, which could then be used for a posterior predictive check (see Model Comparison below).

Also, the authors claim that participants in PGGs often free-ride on the last trial, because they don't have to worry about the consequences for future interactions. This was the stated reason for excluding the last trial from the model and in the fMRI analysis. However, by judging the data in Figure 1D this doesn't seem to be the case as there is no sudden drop in contribution on the last trial.

Computational Modeling

The top-level computation of the model is a weighted linear combination of the relative individual utility of contributing vs free-riding and the group utility that represents the cumulative expected reward for the remaining trials (p. 5). However, Eq 10 defines the group utility not as a cumulative sum of the reward, i.e. the reward does not appear in the summation. There may be just a term missing here, but if the group utility was computed according to Eq 10, then it does not represent what the authors are claiming. Alternatively, if the group utility was changed into a cumulative sum of expected rewards, then likely all results are going to change (modeling, model comparison, and fMRI). Finally, unlike the definition of the individual utility, the group utility is computed without taking the individual action (C/F) into account. I would suggest to make G contingent upon the individual action, which might also help to resolve an inconsistency in the interpretation of the PPI (see below).

The social prediction error (SPE) is updated with a dynamic learning rate that consists of a constant part (α) and a weighted reward prediction error (RPE) (Eq 12). The authors claim that because the RPE-weight (θ) is significantly different from zero, this demonstrates that this weighted RPE (and

hence the dynamic learning rate) is an essential part of the model. However, this claim is overstated and needs to be tested in a formal model comparison (see below) against a model with a constant learning rate.

Overall, there were quite a few careless errors in the model equations:

- (a) indices (e.g. i) are included in some equations and dropped in others,
- (b) Eq 2 is not correct and should read: $p(C) = \text{logit}(Q) = \text{logit}(\omega * I + (1-\omega) * G)$
- (c) in some equations it makes more sense to sum until N and not $N-1$,
- (d) in Eq. 4 it would be better to include k and reference Γ to Eq 11,
- (e) in Eq 15 it would be better to use a different letter than Q as it is easy to confuse with the Q from the social learning model. Also, k is not properly placed.

I would strongly urge the authors to carefully review their equations prior to a resubmission.

Model Comparison

The computation of the BIC values does not adhere to standards and current practice, i.e. the final term ($\ln(N)$) is usually not divided by N . Such a deviation from common practice can change the result of the model comparison itself. It also makes the BIC values unusually low, which also suggests that they were averaged across cross-validation samples and subjects. (The text in the methods suggests that 12-fold cross validation was carried out only once, but I assume in favor of the authors that they iterated across all possible folds in their cross-validation analysis.) A more common approach would be to sum up the log-likelihoods across cross-validation samples and subject and then include these numbers in the penalty terms of the BIC.

Furthermore, the difference in BIC values in Table 1 appear rather marginal, sometimes even just after the decimal. I am not convinced that the best-fitting social learning model is really providing a better fit to the data than the competing models. In that respect, it would be better to convert the BIC values in Bayes Factors (or Bayesian model weights or exceedance probabilities) that would clearly indicate (irrespective of how information criteria were calculated) whether one model has a substantial advantage over the others.

Even if we were to believe that the social learning model was superior, we have no idea whether the model actually fits the data, as there is no posterior predictive check provided in the paper. The authors should simulate new data using their fitted parameter value and then analyze them in a model-free way (e.g. as they did with their initial behavioral analysis). If done this way, it will become clear if the model is able to generate synthetic data that is comparable to the empirical data from the experiment.

Finally, the entire model comparison needs a more systematic approach. The social learning model has many different parameters and their necessary role in the final model should be evaluated using model comparison. In addition, in the supplement, the authors estimate a model with 2 ω parameter and 2 λ parameters, but this variant of the model was never formally tested against the others.

Model-based fMRI

Some of the GLMs are constructed with experimental regressors that are never tested. For instance, in GLM1 there are regressors for the outcome in trial t and decision phase in trial $t+1$, but all model-

based signals (individual and group utility are modeled on the outcome. Why? It is common practice to model value signals at the time of the decision, not the outcome. If the goal was to look for value computation immediate after the outcome, then the utility of trial $t+1$ should have been modeled at the outcome of trial t , and the regressor at the decision phase is superfluous.

Furthermore, the authors state that the individual and group utility are independent - an important information if they are to be modeled on the same outcome event. It would be nice to see that they are uncorrelated. Unfortunately, it has become a sort standard practice to create different GLMs to test for different model-based terms. This avoid the problem of multi-collinearity of different model-based regressors and the authors also follow this unfortunate practice here. However, running several different GLMs is just avoiding, not addressing the problem as the interpretation of different GLMs could become obscured if the inherent correlation between signals is not addressed. Therefore, it is even more important to report correlations between different regressors and model-based signals.

It is confusing that GLM2 conflates the dynamic RPE-dependent learning rate with the SPE, which precludes the identification a region coding purely for and SPE signal. Also, this doesn't appear to be a belief update signal, which is rather modeled in GLM3.

The connectivity analysis (psycho-physiological interaction analysis) presented in Figure 4 is meaningless, because the psychological variable is not properly defined. The authors used a simple onset regressor as the psychological modulator, but this is incorrect. A connectivity analysis like this does not reveal any task-specific modulation, but rather general functional connectivity (like in resting state experiments). I suggest that the authors use a physio-physiological interaction, which includes two BOLD time series from different regions as seeds. This seems to be more what the authors are aiming for.

There are some interpretational inconsistencies that are not properly addressed. According to the findings of Figure 1, vmPFC is inversely correlation with the relative individual utility, so it is associated with free-riding. IPFC is positive correlated with the group utility, so it is associated with contributing. However, these contrasting signals are both positively coupled with dmPFC, which the authors claim is coding the strategic decision, for which they are using the action probability of free-riding ($p(F)$). It is unclear, how two opposing signals can be both positively coupled with the action probability for just one of the choices. Furthermore, I think it would make more sense to use the integrated utility Q (Eq 2) instead of the action probability.

Figures

Figure 1D: The t-tests in the middle panel do appear to be corrected for multiple comparisons. Please sue a correction method (e.g. permutation test) to do so.

Figure 2C: The legend states that that these figures show the group utility G , but there is not G in the panels! Also, in the left panel $N-k$ should be above the Σ , not a superscript of Γ , Figures 2B and 2C are not mentioned in the main text

Figure 3B: x-axis show be labeled "low" and "high". It would be better to show the value of θ here, not the product of $\theta * RPE$

Reviewer #2:

Remarks to the Author:

Park et al. have elucidated neural mechanisms underlying human strategic group decision-making. By

using model-based fMRI together with functional connectivity analysis, they have demonstrated that participants' behavior in Public Goods Game was driven by individual utility encoded in vmPFC and group utility encoded in FPC; and that these computational variables were integrated in dmPFC. I appreciate their efforts to conduct the great experiments and the data analyses, and believe that this study could potentially provide significant insights into wide-range of researchers who are interested in human social cognition and decision-making.

My primary concern is about decision-algorithm of other agents. In strategic interactions, scanned participants' behavioral pattern would highly depend on the other agents' behavioral pattern. The authors have claimed "The computer agent was programmed ... in an ecological manner."; however I could not find any justification or validation. That's a critical point. I believe they need to conduct an additional behavioral experiment involving real interactions among human participants, and to show that their main computational model provides the better fit to the data compared with other models do.

My secondary concern is about computational models. The main model looks plausible, but I believe this should be compared with many other models. For example, they can construct a model with $\Lambda = 0$, $\Pi = 0$, a mixed strategy model ($P(C) = p$ where p is a free-parameter), the optimal mixed strategy model ($P(C) = p^*$ where p^* is the optimal probability predicted by the Nash equilibrium) and a model relying only on Group-utility model etc. Furthermore, is it possible to construct a hybrid model of social learning and inequity aversion?

The authors have assumed participants updated their belief about others' decision in a model-free manner. I believe they need to provide justification. As far as I know, many researchers believe that model-based learning is required to predict others' behavior in strategic interactions (e.g., Yoshida et al., 2010).

In the model comparison procedure (see P.5), why did the authors calculate BIC based on cross-validated likelihood? That's strange. If they employed cross-validation to compute likelihood, they can compare likelihood (not BIC). If they employed BIC, likelihood should be derived without cross-validation. I believe cross-validation is not valid in this study, as each data (i.e., trial) in the repeated-game experiment is not independent from one another. I would recommend Bayesian model selection (Stephan et al., 2009) or hierarchical modeling approach (Daw, 2009: <http://www.princeton.edu/~ndaw/d10.pdf>).

In the fMRI analyses, why did the authors focus on feedback phase, not decision phase? What happens if they look into neural correlates of the key computational variables in decision phase?

In the fMRI analyses, why did the authors have four separate GLMs? Is there specific reason? In principle, I believe all the computational variables of interest should be included into one single GLM to evaluate the explained-variance of each variable.

The present study have shown that key computational variables in the decision-making were integrated in dmPFC. To my knowledge, it is still controversial in which brain regions multiple computational variables are integrated for value-based decision-making. Some studies supported the possibility that value integration occurs in dmPFC including dACC (Hare et al., 2011; and Suzuki et al., 2015), while others provided the evidence for value integration in vmPFC (Behrens et al., 2008; Hare et al., 2010; Smith et al., 2014; and Lim et al., 2013; Suzuki et al., 2017). It would be interesting to discuss this issue in the Discussion section.

Why was the number of interaction fixed ($T = 15$)? A conventional way in this type of repeated-game

experiments is that the number of interaction is determined stochastically. The concept of "backward induction" in Game Theory predicts that participants do not cooperate in any trials (not only the last trials!) in this type of finite repeated interactions.

In Figure 4A, the activation labeled ACCg is corpus callosum?

The authors said they recruited N strangers for each experiment. How did they confirm the participants are strangers?

As far as I understand, when decision-making participants could see information about k , t and $T-t$. How about including these variables into GLM as regressors of no-interest?

Reviewer #3:

Remarks to the Author:

This study by Park and colleagues investigated brain mechanisms underlying the computations involved in social decision-making during public-good game. In the experiment, participants played multi-round, threshold public good game with different groups of 4 virtual people. Authors reported that participants' decision could be modeled as a combined function of individual and group utilities, both of which depended on participants' learned estimate for the number of free-riders among other people at a given round. Using fMRI, authors also reported that activations in separable regions of the brain (ventromedial prefrontal cortex, lateral frontopolar cortex, dorsomedial prefrontal cortex) were correlated with modeled decision variables such as utilities and estimated choice (contribution vs. free-riding) probabilities at a given trial.

Authors used computational approach to model the latent decision variables that allows authors to quantitatively characterize behavior and to look for neural correlates for the latent variables, which is the strength of the study. The public good game in group might provide a good platform to study the mechanisms underlying complex reasoning processes involved in social decision-making.

However, behavioral analyses are rather underwhelming, which makes it hard to evaluate how well the best model captures the representative behaviors of participants during the game. Analyses of fMRI data (GLM models) are also underwhelming and didn't take account the variables whose effects could be potentially confounded with the main results reported in the study. Descriptions of some analyses lack of clarity. In summary, additional behavioral and neural analyses that can convincingly support the validity of the computational model as well as main results from fMRI data could improve the significance of the study.

Major comments:

1. Authors need to present the results of behavioral analyses as well as how well computational model fits the main aspects of participants' behavior. As an example of the most fundamental analyses, authors could use logistic regression model and see how participants' choice (contribution vs. free-ride) at a given round was influenced by relevant variables, such as number of free-riders (among other people; n_F) and participant's choice in the past few trials, trial history of reward, success/failure of producing public good, interaction of reward and choice (i.e. win-stay-lose-switch), etc. to name a few. To see if " $N-k$ " (among $N-1$) is the critical value that determines participants' choice, separate regressors for $n_F=N-k$, $n_F<N-k$, $n_F>N-k$. If participants' behavior is consistent with the model prediction, regressor for $n_F=N-k$ should have positive coefficients (i.e. participants tend to contribute when $n_F=N-k$ in the previous trials compared to other n_F s), consistent with the modeled effect of

individual utility. On the other hand, the regressor for $nF < N - k$ would provide evidence for the effect of modeled group utility. If group utility is determining factor of choice, then the regressor for $nF < N - k$ should have positive coefficients (i.e. the more other people contributes, the more participant contributes), while negative coefficients indicate stronger effect of individual utility (i.e. the more other people contribute, the less participant contributes). This is only one example. Authors need to provide strong evidence that participants' behavior was consistent with model predictions.

2. Although social-game paradigm provides an opportunity to study the strategic decision (i.e. iterative reasoning, higher-order beliefs on other's beliefs, etc.), it is unclear what are the strategic components of participants' behavior in this study. First, group utility in social learning model might capture it. However, the estimated weight ($w \sim 0.8$) for individual utility is very high, suggesting that individual utility was major determinant of the choice. The BIC difference between social learning model and myopic model seems to be only marginal. In addition, it is not clear whether the model-term group utility was capturing only the temporal decay of contribution tendency over time (i.e. $1 - K^{T-t+1}/1-K$), not the effect of successful group cooperation (i.e. probability that $>k$ people would contribute). Second, participants might have used iterative reasoning, forming higher-order beliefs on other people's beliefs. It may be helpful for the authors to explore if higher-order beliefs (e.g. belief of other people on the belief of other people) can partially explain participants' behavior. Finally, previous studies have shown that dmPFC region can be involved in strategic reasoning or switching/arbitration between different strategies (e.g. Hampton et al., 2008; Seo et al., 2014). It would be helpful, if authors can more clearly describe what are the novel aspects of "strategic decision" that could be studied in public good game in groups, and what are the novel insights readers can gain from this study about the function of dmPFC in strategic decision-making.

3. In the analyses of fMRI data, authors used separate GLM models to look for activations correlated with different sets of decision-variables (e.g. individual/group utilities-GLM1, prediction errors for the belief on others' probability of free-riding (PFR) – GLM 2, PFR itself – GLM 3, estimated choice probability – GLM 4). However, these variables tested in separate models are not necessarily independent of each other. For example, in the social learning model, PFR (GLM 3) and prediction error for PFR (GLM 2) are linearly correlated. Choice probability (GLM 4) is also correlated with individual and group utilities (GLM 1). Therefore, in order to know the effect of each variable independent of other correlated variables, all the co-linear variables need to be included in the same regression model. Otherwise, the significant effect of one variable tested in one model could actually reflect the effect of other correlated variables.

4. In general, the description of GLM models in method section is lack of clarity. Particularly, it is unclear how the PPI analysis was done. Authors need to provide better description of what are the "psychological" variables that modulate the functional connectivity. Including equations would help.

5. Authors argued that dmPFC might be involved in strategic decision, as its activation was correlated with choice probability estimated by the model. However, it is not clear how the activation correlated with choice probability can be the evidence for strategic decision. If the decision based on group utility is a "strategic choice" as opposed to a choice based on individual utility, then the region whose activation was correlated with group utility could be involved in "planning" strategic choice or switching between strategies. The activation correlated with choice probability could be simply related to the execution of final decision. Activation related to the "arbitration" between two strategies (individual vs. group utility) can be also related to strategic choice (Lee et al., 2014). This issue is related to the comment #2. It would be helpful if authors provide clear conceptual framework for what "strategic" choice means in the public good game, as well as valid quantitative measurement of strategic choice.

Reviewer #1 (Remarks to the Author):

Comment 1: Figure 1D show basically the same thing in 3 different panels, namely that participants contribute less in $k=2$ trials and that their probability of contribution declines across the PGG. I like the trial-by-trial display, but it would be more informative to combine these data with the success of the subjects (just as in the companion paper using POMDPs (Khalvati et al., <https://www.biorxiv.org/content/early/2018/09/17/419515>). This could reveal an interesting pattern in the model-free behavior, which could then be used for a posterior predictive check (see Model Comparison below).

Answer to the comment 1:

In our revised version of the paper, we have now removed the previous panels C left and right, D and E. We kept the panel showing the mean contribution rate across trials, which provides unique information to the readers about the changes in contribution rate during the public goods game. In addition, we now include the model-free analysis that explains the decision on trial t based on one's previous decisions and outcomes from trials $t-1$ to $t-3$ (current panel **D**). Moreover, we now also provide the formal model comparisons between model-based analysis and model-free analysis using Bayesian model comparison, supporting that the strategic decisions in the current experiment are driven by model-based behavior (current Panel **E**; See our answer to the comment 6). Last, we tested the contribution of each of free-parameters for predicting contribution decision using a cross-validation approach (current Panel **F**; See our answer to the comment 4).

Comment 2: Also, the authors claim that participants in PGGs often free-ride on the last trial, because they don't have to worry about the consequences for future interactions. This was the stated reason for excluding the last trial from the model and in the fMRI analysis. However, by judging the data in Figure 1D this doesn't seem to be the case as there is no sudden drop in contribution on the last trial.

Answer to the comment 2:

We agree with the reviewer. In the revision, in our behavioral and fMRI analyses, we have now included decisions on every trial, including the last trial. Therefore, the number of data points increased from 168 to 180 per participant (15 trial \times 12 Public goods games). We have

made the following changes to the Methods section, on p.14 to indicate that all the trials are now included in the analysis:

Methods (p.15)

The goal of the computational model is to predict the contribution decision at trial t while a participant plays a threshold PGG with repetition ($t = [1:15]$).

Comment 3: The top-level computation of the model is a weighted linear combination of the relative individual utility of contributing vs free-riding and the group utility that represents the cumulative expected reward for the remaining trials (p. 5). However, Eq 10 defines the group utility not as a cumulative sum of the reward, i.e. the reward does not appear in the summation. There may be just a term missing here, but if the group utility was computed according to Eq 10, then it does not represent what the authors are claiming. Alternatively, if the group utility was changed into a cumulative sum of expected rewards, then likely all results are going to change (modeling, model comparison, and fMRI). Finally, unlike the definition of the individual utility, the group utility is computed without taking the individual action (C/F) in account. I would suggest to make G contingent upon the individual action, which might also help to resolve an inconsistency in the interpretation of the PPI (see below).

Answer to the comment 3:

There seems to be a misunderstanding because we did include the reward R in our definition of the group utility G_t . The R is a multiplicative factor of the cumulative sum over the belief about the probability that l people free-ride among $N - 1$. The cumulative rewards for the remaining trials can be defined with another \sum sign and this part equals the outer part of the first \sum in the group utility model. This misunderstanding might have been caused by the fact that we did not include the unfolded formula but the simpler version of the equation. We appreciate that it may be helpful for readers to see the unfolded formula before the simpler version. To clarify this point, we now include the cumulative reward in the new version of the manuscript (p. 18) as follows:

$$G_t = \sum_{j=t}^T RK^{T-j} \sum_{i=0}^{N-k} \Gamma_t^i$$

$$= \frac{1 - K^{T-t+1}}{1 - K} R \sum_{i=0}^{N-k} \Gamma_t^i \quad (8)$$

Therefore, the \sum sign in the previous manuscript in the group utility model indicated the possible scenarios of other players' decisions in which the group can generate public goods. Therefore, it comprises all possible cases in which the group can generate the public goods given that one contributes. It includes the case in which there are no free-riders among others ($i=0$) to the case in which there are $N-k$ free-riders ($i=N-k$). The case in which the group won't generate the public goods can be omitted since the allocated expected utility is 0 MU.

Comment 4: The social prediction error (SPE) is updated with a dynamic learning rate that consists of a constant part (alpha) and a weighted reward prediction error (RPE) (Eq 12). The authors claim that because the RPE-weight (theta) is significantly different from zero, this demonstrates that this weighted RPE (and hence the dynamic learning rate) is an essential part of the model. However, this claim is overstated and needs to be tested in a formal model comparison (see below) against a model with a constant learning rate.

Answer to the comment 4:

The social learning (SL) model contained 4 free-parameters, α , θ , π , and λ , which denote the learning rate, the weight on the reward prediction error, one's altruistic tendency, and the subjective contribution cost. As the reviewer suggested, we have now tested whether these parameters are necessary to explain strategic decision-making by showing that removing any one of these parameters causes a decrease in the quality of the fit. The quality of fit was assessed by leave-one-block-out cross-validation using -2-log likelihood on test datasets, which allowed us to estimate the contribution of each free-parameter from an independent dataset from the training dataset used for estimating the parameters. In the revision, we reported the changes in likelihood (%) when we fixed each of the free-parameters to a constant compared to the full SL model. To address this point, we included the following paragraph in the Model validation and comparison section in the results on page 6 and **Figure 1 F** :

Model validation and comparison (p.6)

Last, we tested the contribution of free-parameters of social learning models to the quality of the fit. The social learning model contained four free parameters, α , θ , π , and λ ,

respectively associated with the learning rate on the estimated decision of others, the weight on the reward prediction error, one's altruistic tendency, and the subjective contribution cost. We tested whether these parameters were necessary to explain strategic decision-making by investigating whether removing any of these parameters causes a decrease in the quality of the fit. This was assessed with the changes in log likelihood using the same cross-validation procedure (leave-one-block-out). We found that the goodness of fit decreases when fixing any of four parameters as a constant value (**Figure 1F**).

Figure 1 F. Changes in quality of fit resulting from removing free-parameters in the social learning model; α , learning rate; θ , weight on learning rate; λ , contribution cost; π , willingness to make altruistic decisions.

Comment 5: Overall, there were quite a few careless errors in the model equations:

- (a) indices (e.g. i) are included in some equations and dropped in others,
- (b) Eq 2 is not correct and should read: $p(C) = \text{logit}(Q) = \text{logit}(\omega * I + (1-\omega) * G)$
- (c) in some equations it makes more sense to sum until N and not $N-1$,
- (d) in Eq. 4 it would be better to include k and reference Gamma to Eq 11,
- (e) in Eq 15 it would to use a different letter than Q as it easy to confuse with the Q from the social learning model. Also, k is not properly placed.

I would strongly urge the authors to carefully review their equations prior to a resubmission.

Answer to the comment 5:

Our answers to these points are the following:

- (a) In the formula, i was not an index. It should be distinguished from I_t which indicates the individual utility or One in $\mathbb{E}(C_t)_{One}$ in Eq. 2. As we mentioned in the methods section, i is the number of free-riders among others. We are afraid that reviewer #1 thought that i was the index of the participant herself (we used “ One ” as the index for that purpose, instead). If it is the case,

he/she misread our computational model. To avoid such misunderstanding, this is now explicitly mentioned in our revised manuscript on page 16.

Methods (p.16)

Furthermore, the expected reward of contribution ($\mathbb{E}(C_t)$) and that of free-riding ($\mathbb{E}(F_t)$) allocated to the participants (called ‘one’) are determined by their belief about Γ^i which was defined as the probability that the number of free-riders among $N-1$ others will be i at round t (i is the number of free-riders among others; $0 \leq i \leq N-1$).

(b) We thank the reviewer for spotting this mistake. We have now corrected it in equation 1 (p.15):

$$\begin{aligned} p(C_t) &= \text{logit}(Q_t) \\ Q_t &= \omega I_t + (1 - \omega)G_t \end{aligned} \tag{1}$$

(c) Since the model assumes that participants simulate the decision of other individuals except herself, $N - 1$ is correct.

(d) We do not need to include k in Γ because the way R_C and R_F are defined, they already account for k . Indeed, since Γ^i indicates the probability that the group has i free-riders among others ($N - 1$) except the participant herself, “ $\sum_{i=0}^{N-1} \Gamma^i$ ” in Eq. 4 indicates the probability (p) of all possible interactions. Therefore, “ $\sum_{i=0}^{N-1} \Gamma^i (R_C)$ ” or “ $\sum_{i=0}^{N-1} \Gamma^i (R_F)$ ” include the sum of expected utility when $i = 0, i = 1, i = 2, i = 3$ and $i = 4$, because participants always played with four other individuals, in this experiment, i can be any integer value from 0 to 4. Moreover, because (R_C) and (R_F) respectively indicate the reward allocated to a participant when she contributed or when she made a free-riding decision, and were inherently defined as variables (Eq. 6), we do not need to include the decision threshold k to define expected utility of participants in Eq. 3 (Eq. 4 in the previous manuscript). Γ indicates the probability of the group, and Γ^{N-k} Eq. 11 should be read in association with Eq.9 (individual utility) rather than Eq.3.

$$\mathbb{E}(C_t)_{one} = \lambda + \sum_{i=0}^{N-1} \Gamma^i (R_C)$$

$$\mathbb{E}(F_t)_{One} = \sum_{i=0}^{N-1} \Gamma^i(R_F) \quad (3)$$

$$R_F = \begin{cases} R & \text{if } (N-1) - i \geq k \\ 0 & \text{if } (N-1) - i < k \end{cases}$$

$$R_C = \begin{cases} R & \text{if } (N-1) - i \geq k-1 \\ 0 & \text{if } (N-1) - i < k-1 \end{cases}$$

$$R_C - R_F = R \text{ when } i = N - k \text{ (} R_C \neq R_F \text{)}$$

$$R_C - R_F = 0 \text{ otherwise (} R_C = R_F \text{)}$$

(6)

(e) We thank the reviewer for pointing out this mistake, which we have now corrected in equation 15 (p.20):

$$p(C_t) = p(C_1) \times p(W_t)$$

$$\text{where } p(W_t) = \text{logit} \left(\kappa \sum_{i=1}^{t-1} \varepsilon R_i - \delta(C_i - \bar{C}_i) \right) \quad (15)$$

Comment 6: The computation of the BIC values does not adhere to standards and current practice, i.e. the final term (ln(N)) is usually not divided by N. Such a deviation from common practice can change the result of the model comparison itself. It also makes the BIC values unusually low, which also suggests that they were averaged across cross-validation samples and subjects. (The text in the methods suggests that 12-fold cross validation was carried out only once, but I assume in favor of the authors that they iterated across all possible folds in their cross-validation analysis.) A more common approach would be to sum up the log-likelihoods across cross-validation samples and subject and then include these numbers in the penalty terms of the BIC. Furthermore, the difference in BIC values in Table 1 appear rather marginal, sometimes even just after the decimal. I am not convinced that the best-fitting social learning model is really providing a better fit to the data than the competing models. In that respect, it would be better to

convert the BIC values in Bayes Factors (or Bayesian model weights or exceedance probabilities) that would clearly indicate (irrespective of how information criteria were calculated) whether one model has a substantial advantage over the others.

Answer to the comment 6:

We now provide the sum of BIC as suggested by reviewers, not divided by the number of samples, N . In addition to the BIC, we now also provide the exceedance probabilities computed from Bayesian model selection (BMS). Both measures indicate that the social learning model still outperforms the other models. The protected exceedance probabilities are shown in Fig. 1F, showing that the social learning (SL) model outperforms the alternative models (Myopic: M, Group utility: GU, and inequity aversion: IA models) and model-free (MF) prediction (See our response to the comment 7 for details about MF analysis). We have now added the following analysis on page 5 to answer this comment:

Model validation and comparison (p.5)

*To test whether the social learning model captures the characteristics of decisions during PGG, we performed a number of analyses. First, we fitted those four computational models to the participants' actual choice data. Using the Bayesian information criteria (BIC; **Equation 16**) which penalizes additional free parameters, we compared the goodness of fit of each model (**Table 1**). We found that the social learning model better explained participants' decision during PGG than other alternative models, and this was also true when comparing the posterior model probabilities using Bayesian model selection (BMS, **Figure 1E**). ...*

Taken together, these analyses show that behavior in the volunteer's dilemma can be best captured by the social learning model. According to this model, people compute the following key variables: individual utility, group utility, their integration, a prediction of the group's likely choice, and a corresponding social prediction error. Next, we harnessed quantitative predictions from the social learning model to identify the neural correlates of these computations.

Figure 1 E. Model comparisons based on the Bayesian model selection. The protected exceedance probabilities indicate that the social learning (SL) model explains decisions in PGG better than other alternative models: Myopic (M); Group utility (GU); Inequity aversion (IA); Model-free analysis (MF).

Comment 7: Even if we were to believe that the social learning model was superior, we have no idea whether the model actually fits the data, as there is no posterior predictive check provided in the paper. The authors should simulate new data using their fitted parameter value and then analyze them in a model-free way (e.g. as they did with their initial behavioral analysis). If done this way, it will become clear if the model is able to generate synthetic data that is comparable to the empirical data from the experiment.

Answer to the comment 7:

We thank the reviewer for this important comment. We now provide evidence that the model-free characteristics in actual data are captured also in the synthetic data generated by the computational model. This new analysis also allows us to address the point of the reviewer. In the revised manuscript, we provide the results of the model-free analysis for both actual and model generated data. This analysis uses multiple logistic regression analysis to predict the decision at trial t with the number of free-riders (nF), the previous decision (D), the success or failure (S/F) to generate the public goods, and the win-stay and loose-switch strategy (Ws/Ls). We also included these regressors up to $t-3$ trials into the past. To perform unbiased tests, the synthetic data were generated with leave-one-block-out cross-validation with the decision of one public goods game (PGG) made (test set) using the parameters estimated from other 11 PGG (training set). We have clarified this issue by stating that we did perform all possible 12-fold repetitions to generate synthetic data the same size as the actual data. The results of this model-free analysis show that the social learning model generally captures the model-free characteristics.

We have made the following changes to the results section to describe in detail this model-free analysis on page 3 and the model validation and comparison on page 4:

Model-free analysis (p.4)

Using a mixed effect logistic regression model, we examined how participants' contribution decision at a given round, t was influenced by relevant variables, such as the number of free-riders (nF), the previous decision (D), the success or failure (S/F) to generate the public goods, and the win-stay and loose-switch strategy (Ws/Ls). We also included these regressors up to $t-3$ previous trials. We found that participants were not contributing more when the number of other free-riders increased ($t_{24} = 1.73, p=0.10$), nor when there was success in generating the public goods ($t_{24} = -0.30, p=0.77$). This result suggests that participants generally used a model to make a strategic decision rather than simply repeating their decisions that generated the public goods in previous interactions in a model-free way (**Figure 1D**).

Figure 1 D. Model-free analysis. We regressed the behavioral decision on the number of free-riders (nF), decision (D), success or failure to generate the public goods (S/F), and win-stay and lose-switch strategy (Ws/Ls) in previous trials up to three trials back. (mixed effect logistic regression).

Model validation and comparison (p.5)

Given that the social learning model outperforms the alternative models, we tested further whether the social learning model accurately predicts the series of decisions made during a PGG (test-set) from the independent data (training-set). To do this, we conducted a leave-one-block-out cross-validation approach such that the decision made for the N -th PGG is predicted based on the parameters estimated by fitting the model to decisions made for the other 11 PGGs except for the N -th PGG of each participant. This process was repeated 12

times. In doing so, we simulated new data using our parameters fitted to an independent dataset. First, the social learning model predicts that for the PGG with stronger volunteer's dilemma, participants are less likely to contribute to the PGG compared to the weaker volunteer's dilemma across trials. The mean contribution rates in the model-predicted dataset are shown in **Figure S1 B**. Second, we also performed a model-free analysis of the decisions generated by computational models. We found that the model free characteristics that we observed in the actual behavior were largely recapitulated in the model-generated behaviors (**Figure S1 C**).

Figure S1 B. Average probability to contribute in each round ($2 \leq t \leq 15$) generated by the computational model. As in the actual decisions made by human participants, the weaker the volunteer's dilemma ($k=4$), the higher rate of contribution decisions was made by the social learning model during the PGG. Error bars indicate s.e.m; *: $q < 0.05$ FDR corrected for multiple comparisons. **C.** Model-free analysis of the synthetic data. We regressed the behavioral decision on the number of free-riders (nF), previous decision (D), success or failure to generate the public goods (S/F), and win-stay and lose-switch strategy (Ws/Ls) in previous trials up to three trials back. Error bars indicate s.e.m; *: $p < 0.05$, **: $p < 0.01$ (mixed effect logistic regression).

Comment 8: Finally, the entire model comparison needs a more systematic approach. The social learning model has many different parameters and their necessary role in the final model should be evaluated using model comparison. In addition, in the supplement, the authors estimate a model with 2 omega parameter and 2 lambda parameters, but this variant of the model was never formally tested against the others.

Answer to the comment 8:

Please see our answers to the comment 4 and to the comment 6, where we took a more systematic approach and fitted four computational models to the participants' actual choice data using the Bayesian information criteria. We discarded the results testing potential changes in parameters (ω and λ) in the revised manuscript. We decided to be cautious, as the reviewer pointed out that it is hard to control covariance between free-parameters if we assume that participants may use different parameters in different levels of volunteers' dilemma conditions, especially since this would only use half of the dataset for different levels of volunteer's dilemma to fit the model.

Comment 9: Some of the GLMs are constructed with experimental regressors that are never tested. For instance, in GLM1 there are regressors for the outcome in trial t and decision phase in trial $t+1$, but all model-based signals (individual and group utility are modeled on the outcome. Why? It is common practice to model value signals at the time of the decision, not the outcome. If the goal was to look for value computation immediate after the outcome, then the utility of trial $t+1$ should have been modeled at the outcome of trial t , and the regressor at the decision phase is superfluous.

Answer to the comment 9:

We believe that the reviewer understood that the decision phase was modeled with parametric modulators. However, it was not. We modeled the decision phase without parametric regressors to control for other cognitive process (because this decision phase did occur on each trial and has to be modeled to explain this variance) but not to identify the brain signals encoding computational variables at the time of decision.

Moreover, as the reviewer suggested, we added the results of the alternative GLM (**Figure S6**) which modeled the brain signals at the time of the decision to examine the neural encoding of specific computational variables. Results from this alternative time point revealed weaker effects in a similar network of areas including the vmPFC and IFPC. As in previous studies using repeated decision tasks, we find that more variance is captured at the preceding feedback, most likely because subjects can already make their next decision since all the relevant information for the upcoming decision is in hand.

In the current study, we are only predicting the decision of the following trial, t based on the brain signals elicited during the feedback phase of the previous trial, $t-1$. Otherwise, as the reviewer pointed, it is "surplus". This is now described in a clearer fashion in the revised manuscript (see our changes below). Several published studies have shown that, during social

interactions, the brain signals at the time of feedback of the previous trial are predictive of the decision of the following trial¹⁻⁶. We found that this is also true in our study. It is also important to note that, in the current design of the task, participants should rate the level of satisfaction after getting feedback but before deciding in the next trial. This additional temporal dissociation between feedback and decision for the next trial may weaken the predictability of the brain activity measured at the time of the decision phase. To resolve this issue, in the new version of the manuscript, we include such secondary analysis in the supplementary results on page 7 showing an attenuation of brain signals computing computational variables at the time of decision. Moreover, we found that the level of difficulty in decision-making was not correlated with the reaction times, which supports the notion that the decision may have already been formed before the decision phase.

Results (p.7)

*Previous studies indicate that during repeated social interactions in groups, individuals are more likely to update their belief when observing the decision of others at trial $t-1$, which explains variance in decision-making at trial t ^{1,3,7}. For this reason, we modeled the brain responses at the outcome phase, i.e. the utility of the decision at trial t was modeled at the time of receiving the outcome of social interactions at trial $t-1$. To test the alternative hypothesis that the computational variables are encoded at the time of decision-making, we also analyzed the fMRI data at the time of decision-making phase. The brain responses were modeled in the same way as in GLM1, except that we modeled brain responses at the decision phase on trial t to predict the decision on trial t . We found that activity in the vmPFC $(x,y,z)=(0,59,1)$ inversely correlated with the model estimated I_t , and that activity from the IFPC $(x,y,z)=(39,44,1)$ correlated positively with G_t ($p<0.001$, uncorrected). These activations were not significant at the whole-brain FWE corrected at cluster level ($p_{FWE} = 0.69$ for vmPFC and 0.79 for IFPC; **Figure S6**). In addition, we found that the model estimates of decision difficulty ($|p(C) - 0.5|^{-1}$) did not significantly explain the reaction times from the decision onset ($t_{24}=-0.16$, $p=0.87$), suggesting that participants were more likely to have made a decision when the outcome of the previous interaction was revealed.*

Figure S6. Neural correlates of Individual utility (I_t) and Group utility (G_t) at the time of decision onset. Activity (in blue) in the ventromedial prefrontal cortex (vmPFC) at the time of decision on trial t inversely correlated with I_t . Activities (in red) in the right lateral frontopolar cortex (IFPC) and bilateral inferior parietal lobule (IPL) at the time of decision on trial t positively correlated with the estimated G_t . The statistical maps are thresholded at $p < 0.005$, uncorrected (darker color). The lighter color map was thresholded at $p < 0.001$, uncorrected.

Comment 10: Furthermore, the authors state that the individual and group utility are independent - an important information if they are to be modeled on the same outcome event. It would be nice to see that they are uncorrelated. Unfortunately, it has become a sort standard practice to create different GLMs to test for different model-based terms. This avoid the problem of multi-collinearity of different model-based regressors and the authors also follow this unfortunate practice here. However, running several different GLMs is just avoiding, not addressing the problem as the interpretation of different GLMs could become obscured if the inherent correlation between signals is not addressed. Therefore, it is even more important to report correlations between different regressors and model-based signals.

Answer to the comment 10:

To resolve the potential multicollinearity issue raised by the reviewer, we have now changed the GLMs to deal with this potential problem. We ran new GLM analyses in which we used the following regressors: estimates of the individual utility (I), the group utility (G), and one's belief about the decision of others (γ). Moreover, we now allow these regressors to compete with other regressors: the weighted prediction errors (wPE), the magnitude of reward (R), and the current trial (t).

Importantly, we examined the brain signals encoding I and G (using GLM1), and γ (using GLM2) while controlling for the covariation between regressors. To do this, we regressed out

the shared variance with the regressors of interest by performing a partial correlation before we inputted the other regressors.

Specifically, for GLM1, which serves to identify the brain activity encoding individual utility (I) and group utility (G), we included the parametric regressors of utilities (I and G), as well as wPE^{IG} , R^{IG} , t^{IG} and γ^{IG} . For instance, the parametric regressor, wPE^{IG} was computed as $wPE - I\beta_{wPE,I} - G\beta_{wPE,G}$ where $\beta_{wPE,I}$ and $\beta_{wPE,G}$ determine to what extent wPE was explained by I and G (partial correlation coefficient). This process of partialling out covariance between regressors was also applied for computing other regressors, R^{IG} , t^{IG} and γ^{IG} . This partial correlation allowed us to control for the confounding variable of other regressors (wPE , R , t and γ) by preferentially assigning covariance to regressors of interests without any transformation (I and G), which is done by computing the partial correlation coefficients. Importantly, wPE^{IG} , R^{IG} , t^{IG} , and γ^{IG} are still highly correlated with wPE , R , and γ respectively, while they do not correlate with I and G anymore. In doing so, we are able to identify brain activity specifically correlating with I and G , while partialing out the signals which were more likely modulated by other regressors of non-interests. This partial correlation method allows us to prioritize multiple regressors of interests equally (I and G in GLM1) and to preserve their values while competing with other regressors of no interest (wPE , R , and γ in GLM1). The advantage of this method over the classical orthogonalization method is that the latter would only allow us to keep one regressor's value and to change the values of sub-rank regressors sequentially according to their priority.

For GLM2, which serves to identify brain activity specifically encoding γ , we used the same parametric regressors as for GLM1. In GLM2, I^γ , G^γ , wPE^γ , R^γ , and t^γ were included while partialing out their confounding variable with γ . As described above, I^γ was computed as $I - \gamma\beta_{I,\gamma}$ where $\beta_{I,\gamma}$ indicates to what extent I was explained by γ . In this way, again, I^γ , G^γ , wPE^γ , R^γ , and t^γ are still highly correlated with I , G , wPE , R and t respectively, while they do not correlate with γ anymore. We thus controlled the confounding variables of other regressors (I^γ , G^γ , wPE^γ , t^γ , and R^γ) by prioritizing the effects of regressor of interest (γ). The impact of partial correlation, which resolves the issue of multicollinearity, is shown in the supplementary Figure S5 (see below). We added the following paragraphs to the fMRI data analysis on p.22 :

fMRI data analysis (p.22)

We ran general linear model (GLM) analyses to identify which brain regions encode the following computational variables: estimates of the individual utility (I), the group utility (G), and

one's belief about the decision of others (γ). These computations are serving to make decision at trial, t . In addition, we allow these regressors to compete with other regressors which are serving to process the outcome of the previous interaction, $t - 1$: the weighted prediction errors (wPE , the update term in **Equation 10**) and the reward allocated to the participant (R). To control for the number of remaining trials, we also inputted the trial number, t as an additional regressor. First, we examined the brain signals encoding I and G (GLM1), and examined the brain regions computing γ (GLM2) while controlling for the covariation between regressors. To deal with the multicollinearity issue (**Figure S6 A**), we inputted the regressors of interest (I and G for GLM1 and γ for GLM2), and we inputted the other regressors after regressing out their shared variance with the regressors of interest by performing a partial correlation.

Specifically, for GLM1, which serves to identify the brain regions encoding individual utility (I) and group utility (G), we included the parametric regressors of utilities (I and G), as well as wPE^{IG} , R^{IG} , t^{IG} and γ^{IG} . The parametric regressor, wPE^{IG} was computed as $wPE - I\beta_{wPE,I} - G\beta_{wPE,G}$ where $\beta_{wPE,I}$ and $\beta_{wPE,G}$ indicate to what extent wPE was explained by variances of I and G , which was also applied for computing R^{IG} , t^{IG} and γ^{IG} . This partial correlation allowed us to control for the confounding variable of other regressors (wPE^{IG} , R^{IG} , t^{IG} , and γ^{IG}) by preferentially assigning covariance to regressors of interests without any transformation (I and G). Importantly, wPE^{IG} , R^{IG} , t^{IG} , and γ^{IG} are still highly correlated with wPE , R , t , and γ respectively (See orange colored area in **Figure S6 B**), while they do not correlate with I and G anymore (See purple colored area in **Figure S6 B**). This partial correlation method allows us to prioritize multiple regressors of interests equally (I and G in GLM1) and identify brain activity specifically correlating with each of computational variables while controlling their covariance with other regressors (wPE , R , t , and γ in GLM1). The advantage of this method over the classical orthogonalization method is that the latter would only allow us to keep one regressor's value and to change the values of sub-rank regressors sequentially according to their priority.

For GLM2, which serves to identify brain activity specifically encoding γ , we included γ , I^γ , G^γ , wPE^γ , t^γ , and R^γ . As described above, I^γ was computed as $I - I\beta_{I,\gamma}$ where $\beta_{I,\gamma}$ indicates to what extent I was explained by γ . In this way, again, I^γ , G^γ , wPE^γ , t^γ , and R^γ are still highly correlated with I , G , wPE , t , and R respectively (See orange colored area in **Figure S6 C**), while they do not correlate with γ anymore (See purple colored area in **Figure S6 C**). We thus controlled the confounding variables of other regressors (I^γ , G^γ , wPE^γ , t^γ , and R^γ) by prioritizing the effects of regressor of interest (γ).

The participant-specific design matrices contained the boxcar functions of outcome presentation (from its onset with 4 s duration) from the 1st to the 14th rounds to examine brain activity involved in decision-making for the 2nd to the 15th trials of PGG. Additional regressors of non-interests were as follows: a stick function for the button press onsets and the decision onsets; a boxcar function of error message presentation - trials in which subjects or their team member did not respond -, which was modeled as separate regressors (3 s). In addition, the motion parameters produced by head movement were also entered as additional regressors of no interest to account for motion-related artifacts.

Figure S5. We ran general linear model (GLM) analyses to identify the brain regions encoding the following computational variables: estimates of the individual utility (I), the group utility (G), and one's belief about the decision of others (γ). In addition, we allow these regressors to compete with other regressors: the reward allocated to the participant (R), the weighted prediction errors (wPE), and the trial number (t , to control the effects of the number of remaining trials). **A.** The mean cross-correlation among these regressors. **B.** For GLM1, to deal with multicollinearity of other regressors with the regressors of interests (I and G), we inputted the other regressors between regressors of non-interests (wPE , R , t , and γ) after regressing out their covariance with the regressors of interest by performing a partial correlation. The off-diagonal triangle shows the mean cross-correlation among the regressors inputted into the GLM1 – I , G , wPE^{IG} , R^{IG} , t^{IG} and γ^{IG} . Importantly, these regressors are still highly correlated with their original values (the diagonal highlighted in orange), while they do not correlate with I and G anymore (off-diagonal highlighted in purple). **C.** For GLM2, to deal with the multicollinearity of other regressors with the regressors of interests (γ), we inputted the other regressors between non-interests regressors (I , G , wPE , R , and t) after regressing out their covariance with the regressors of interest by performing a partial correlation. The off-diagonal triangle shows the mean cross-correlation among the regressors inputted into the GLM2 – I^γ ,

G^Y , wPE^Y , R^Y , t^Y and γ . Importantly, these regressors are still highly correlated with their original values (the diagonal highlighted in orange), while they do not correlate with γ anymore (off-diagonal highlighted in purple).

Comment 11: It is confusing that GLM2 conflates the dynamic RPE-dependent learning rate with the SPE, which precludes the identification a region coding purely for and SPE signal. Also, this doesn't appear to be a belief update signal, which is rather modeled in GLM3.

Answer to the comment 11:

We agree with the reviewer about this point. By definition, the prediction error is updated by the monetary prediction error and by the social prediction error. It is therefore difficult to specify whether the neural underpinnings of the weighted prediction errors (wPE) is specific or not to the social prediction error (i.e. to what extent one's expectation about the decision of another player in the group is violated). In the revised manuscript, we have now taken off this result. Instead, we now focus on the neural correlates of γ using the GLM in which the covariance with wPE is taken care by regressing out its partial correlation (see the answer to the comment 10).

Comment 12: The connectivity analysis (psycho-physiological interaction analysis) presented in Figure 4 is meaningless, because the psychological variable is not properly defined. The authors used a simple onset regressor as the psychological modulator, but this is incorrect. A connectivity analysis like this does not reveal any task-specific modulation, but rather general functional connectivity (like in resting state experiments). I suggest that the authors use a physio-physiological interaction, which includes two BOLD time series from different regions as seeds. This seems to be more what the authors are aiming for.

There are some interpretational inconsistencies that are not properly addressed. According to the findings of Figure 1, vmPFC is inversely correlation with the relative individual utility, so it is associated with free-riding. IPFC is positive correlated with the group utility, so it is associated with contributing. However, these contrasting signals are both positively coupled with dmPFC, which the authors claim is coding the strategic decision, for which they are using the action probability of free-riding ($p(F)$). It is unclear, how two opposing signals can be both positively coupled with the action probability for

just one of the choices. Furthermore, I think it would make more sense to use the integrated utility Q (Eq 2) instead of the action probability.

Answer to the comment 12:

The reviewer pointed out that the task-specific modulation was not tested in the previous PPI since we have treated all decisions as being made strategically. Moreover, the reviewer 3 (comment 4) recommended that we provide a better definition of 'strategic decision-making' to define the psychological variable more clearly in the PPI analysis.

Taking these comments into account, we are now providing a better description of what are the "psychological" variables that modulate functional connectivity. According to this point, in the revised manuscript, we performed a new functional connectivity analysis replacing the previous one, adopting the reviewer 3's suggestion (comment 5) to define strategic-decisions as arbitration between strategies. More precisely, in the previous PPI, we adopted a broad definition of strategic decisions and we included all decision trials as a strategic decision. However, this previous definition made it hard to distinguish between the neural mechanisms underlying different motives for strategic decisions. In the new PPI included in the revised manuscript, we addressed this issue by narrowing down the definition of strategic decision-making. That is, we now define the psychological factor of strategic decisions as the trials requiring the arbitration between different decisions. Based on this definition, there are two types of strategic decisions. One of the strategic decisions is to switch one's free-riding (Fr) decision to contribution (Co) in favor of the group utility (Fr→Co). Since a strategic contribution can induce future contribution of others, switching to contribution is a strategy which potentially leads to greater rewards in the long-term. The other type of strategic decision is to switch one's contribution decision to free-riding to maximize one's immediate reward (Co→Fr). These decisions of switching to the other strategy are contrasted with the decision of staying with the previous strategy. To incorporate this strategic decision-making as the psychological factor, we estimated to what extent the model's predicted value of contribution decision is changed from one trial to the next ($\Delta Q = Q_t - Q_{t-1}$). Note that we used the integrated utility Q here as suggested by the reviewer (instead of the action probability).

With this new operational definition of strategic decisions, we performed a new PPI analysis (see description below). First, we observed that activity in the anterior cingulate cortex (ACC) and the ventrolateral prefrontal cortex (vlPFC) increased for trials in which one switches strategy compared to trials in which one stays with the same strategy (Figure 4A). We then performed a functional connectivity analysis using the time series extracted from the seed

regions – the ACC and vIPFC – inputted as the physiological factors. The psychological factor includes the changes in the utility of one’s strategy from free-riding to contribution (positive values) and the changes in the utility of one’s strategy from contribution to free-riding (negative values) which was estimated by the model. Finally, we searched for brain regions encoding their psycho-physiological interactions. This new PPI analysis revealed brain areas showing increasing functional connectivity with ACC or vIPFC when participants changed their strategy. Specifically, when participants changed their decision strategy in favor of long-term collective rewards and switching their strategy to contribution, we observed increased functional connectivity between seed areas to the lateral frontopolar cortex (IFPC), which computes the group utility (G) (red blob in Figure 4B), while showing a decrease in functional connectivity with the ventromedial prefrontal cortex (vmPFC), which computes the individual utility (I) (blue blob in Figure 4B). The results also suggest that, when participants changed their decision strategy in favor of immediate rewards and switched their strategy to free-riding, the functional connectivity of seed areas increased with the vmPFC and decreased with the IFPC. These results clarify how the computational variable of individual and group utilities are encoded in distinct brain areas and how they guide the arbitration between different strategic decisions. We added the following paragraphs about the new PPI analysis and results in Results on p.8, Methods on p.23, and Discussion section on p.11 and Figure 4 A and B:

Figure 4. Neural mechanisms of arbitration between different strategies during the Public goods game. **A.** Activity in the right ventrolateral prefrontal cortex (vIPFC) and the anterior cingulate gyrus (ACC) increased when switching one’s decision during the PGG. The statistical maps are thresholded with the same convention as in Figure 2. **B.** Connectivity analyses between the brain regions engaged in the arbitration between different strategies (the vIPFC and the ACC) and the brain areas encoding the individual utility and the group utility. The circles

represent seed regions from which physiological signals were extracted, and colored blobs show the psychophysiological interaction effect. The ventromedial prefrontal cortex (vmPFC; blue), encoding the individual utility, shows a negative correlation with the activity in seed regions modulated by the probability to change one's strategy to contribution ΔQ ($\Delta Q = Q_t - Q_{t-1}$), predicted by the social learning model ($p < 0.05$ small-volume corrected). The right lateral frontopolar cortex (rIFPC; red), which encoded the group utility, shows a positive correlation with the signals in seed regions modulated by ΔQ ($p < 0.05$ small-volume corrected). For illustrative purpose, the statistical maps are thresholded at $p < 0.005$, uncorrected.

Neural mechanisms arbitrating different strategies (Results part, p.8)

One of the strategic decisions in the current study is switching one's decision to contribution away from immediate individual utility in favor of the long-term group utility (which indicates collective future expected rewards allocated to not only others but also the player oneself). Because a strategic contribution can induce future contribution of others, switching to contribution is a strategy which potentially leads to greater rewards in the long-term. The other type of strategic decision is switching one's decision from contribution to free-riding to maximize one's immediate reward. To investigate the neural underpinnings of such strategic decision-making, we examined the brain areas showing increased activity for the trials in which one switches their decision compared to the trials in which one stays with the previous decision^{8,9}. We found that strategy switching at round t is predicted by increased activity in the right ventrolateral prefrontal cortex (vlPFC), ($x, y, z = 39, 26, 13$), and in the ACC, ($x, y, z = 0, 17, 31$) at the time of the outcome of the previous round, $t-1$ (**Figure 4A** and **Table S1A**, $p < 0.05$, GLM3, whole-brain corrected with FWE at cluster level). These areas are therefore likely to be involved in implementing an arbitration between strategies.

To identify the neural mechanism underlying the arbitration between strategies, we examined how the choice probability modulates the interactions between the brain areas involved in switching decisions and the areas involved in encoding the individual utility and the group utility. We hypothesized that brain regions implementing the arbitration between strategies would show enhanced coupling with those areas encoding subjective utilities. To test for this, we conducted a psychophysiological interaction (PPI) analysis. The physiological variables were the brain signals extracted from the brain areas involved in the arbitration between different strategies (ACC and vlPFC) at the time of feedback. The psychological variable was the model prediction of the changes in decision value (ΔQ) as a function of what

extent one is more likely to change their strategy at the trial, t to contribution ($\Delta Q = Q_t - Q_{t-1}$). That is, ΔQ is positive when one is more likely to change the strategy to contribution and negative when one is more likely to change the strategy to free-riding, while it is close to zero when one is more likely to stay the previous strategy. The decision value Q was predicted by the social learning model (**Equation 1**).

We found that the vIPFC and the ACC showed increased functional connectivity with the right IFPC, $(x,y,z)=(30,50,1)$ ($p < 0.05$, small volume corrected), the same region encoding the group utility. We also revealed that the vmPFC $(x,y,z)=(9,50,-8)$, one of the regions encoding the individual utility, showed the opposite pattern of functional connectivity ($p < 0.05$, small volume corrected). That is, increased functional connectivity was found between the IFPC and the brain areas engaged in the arbitration between strategies as a function of the changes in decision value to contribution strategy, while increased functional connectivity was observed between the vmPFC and the brain areas engaged in the arbitration between strategies as a function of the changes in decision value to free-riding strategy (**Figure 4B** and **Table S1B**). Together, these findings suggest that the neural encoding of the group utility and individual utility, as formalized from the social learning model, inform the arbitration between strategies in vIPFC and ACC during social interactions (**Figure 4C**).

Psychophysiological interaction (PPI) analysis (Methods part p.23)

We define the psychological factor of strategic decisions as the trials requiring the arbitration between different decisions. Based on this definition, there are two types of strategic decisions. One of the strategic decisions is to switch one's free-riding (Fr) decision to contribution (Co) in favor of the group utility (Fr \rightarrow Co). Since a strategic contribution can induce future contribution of others, switching to contribution is a strategy which potentially leads to greater rewards in the long-term. The other type of strategic decision is to switch one's contribution decision to free-riding to maximize one's immediate reward (Co \rightarrow Fr). With this definition of strategic decisions, we were able to examine the functional connectivity specific to the decision of switching to free-riding and the decision of switching to contribution compared to the decision of staying with the previous decision.

Using a PPI analysis, we focused on the time of feedback and examined the changes in functional connectivity between the brain regions involved in the arbitration between two strategies and the brain regions encoding the computational variables (the individual utility, I_t ,

and the group utility, G_t). For the PPI, we defined the seed regions of interest (ROIs) as a 8 mm radius spheres centered on the coordinates extracted from the peak voxel of the right ventrolateral prefrontal cortex (vlPFC), $(x,y,z= 39,26,13)$ in MNI coordinates and the anterior cingulate cortex (ACC), $(x,y,z=0,17,31)$ predicting the arbitration between two strategies (GLM3). The physiological variable is therefore the timeseries extracted from a priori ROIs at the time of feedback phase of $t - 1$ trial (the 1st to the 14th rounds) of the PGG. In addition, we defined the psychological factors as the model prediction of the probability to switch one's strategy to contribution. Specifically, it was computed as the difference in the decision value ($\Delta Q = Q_t - Q_{t-1}$) (**Equation 1**). The decision value, Q is predicted from the social learning model. That is, participants tend to switch to contribution strategy when $\Delta Q > 0$, while they tend to switch to free-riding strategy when $\Delta Q < 0$, and they tend to stay to their current strategy when $\Delta Q \approx 0$. The GLM for the PPI analysis therefore contained the following regressors: (1) physiological factors, BOLD signals from the ROIs, (2) psychological factors, and (3) PPI factors, interaction terms of the psychological and physiological factors, as well as the same regressors of no interests that we used for GLM1.

The statistical significance of the functional connectivity in the vmPFC and in the right IFPC were tested within anatomically defined independent regions of interest (ROIs). The ROIs were defined by a previous study that anatomically parcellated the prefrontal cortex according to the resting state connectivity¹⁰. Specifically, we respectively used two parcellations annotated as area '11m' and the 'frontopolar cortex lateral' (FPI).

Discussion (p.11)

In the current study, participants who adopted a mixed strategy might flexibly switch their strategies between free-riding in favor of the individual utility and contribution in favor of the group utility. Considering strategic decision-making as the flexible arbitration between different strategies, we observed that the activity from the vlPFC and the ACC increased when switching between strategies, suggesting that these brain regions compute an arbitration signal in strategic decision making during collective decisions. It is important to note that here, we defined a change in strategic-decision as a change in model predicted decision-value, because it is not possible to dissociate a change in strategy from a change in behavioral response. This interpretation relates to the results of a previous study suggesting that the vlPFC is engaged in controlling model-based and model-free decision strategies⁹. Also, the functional connectivity of the ACC to other brain areas tracking the history of others' decisions and one's own preference

has been shown to guide collective decisions during consensus decision for a group ¹¹. Moreover, the activity of the ACC correlates with individual differences in the degree to which an individual prefers one strategy over the other during competitive decision-making ¹². Moreover, the functional connectivity between the vIPFC and ACC –regions selectively engaged for the event predicting the strategy switch– decreased with the vmPFC encoding ‘individual utility’ and increased with the IFPC encoding ‘group utility’ when the probability of contribution was high at the time of feedback. The increased strength of the relationship between ACC-IFPC may relate to a central role of the dACC-dIPFC interactions proposed in relation to a decision variable called prospective value, as opposed to an immediate myopic value ^{13,14}. In such framework, when making sequential decisions, the overall value of the environment can be decomposed into a myopic component, corresponding to the average benefits that might immediately follow a decision, and prospective value, corresponding to future benefits that might accrue over the longer term by taking a particular choice now ¹³. In light of this previous study, our results suggest that the strength of the dACC-IFPC connectivity increasing with group utility may reflect choice strategy related to prospective valuation.

Comment 13: Figure 1D: The t-tests in the middle panel do appear to be corrected for multiple comparisons. Please use a correction method (e.g. permutation test) to do so.

Answer to the comment 13:

The changes in contribution rates were compared across trials. The method we used for multiple comparisons was the false discovery rate (FDR) to correct at the threshold, $q=0.05$ using the methods introduced by Benjamini and Hochberg ¹⁵. We make this clear this in the figure legend and have also made the following changes in Results (p.4) and Figure 1 (p.30):

Results (p.4)

false discovery rate (FDR)¹⁵ corrected for multiple comparison; **Figure 1C**

Figure 1 C. ... **: $q < 0.01$, *: $q < 0.05$ FDR corrected for multiple comparisons

Comment 14: Figure 2C: The legend states that that these figures show the group utility G, but there is not G in the panels! Also, in the left panel N-k should be above the Sigma, not a superscript of Gamma, Figures 2B and 2C are not mentioned in the main text

Answer to the comment 14:

We thank the reviewer for noting these mistakes, which are now corrected. We have made the following changes to Fig 2B and 2C:

Figure 2 legend (p.31)

B. Conceptual illustration of individual utility (I_t) according to one's belief about the decision of another (γ). After the feedback of the previous trial, $t-1$, a participant may compute the immediate expected utility for oneself (I_t) while computing the cumulative expected rewards for the group (G_t) for remaining interactions ($T-t+1$, where $T=15$). I_t correlates with the binomial probability density function of γ_t which indicates one's belief about the probability that one of other members will free-ride at round t . **C.** Conceptual illustration of group utility (G_t) according to one's belief about the decision of another (γ). G_t depends on the probability that the group generates the public goods and the number of remaining trials. (Left) The probability to generate

the public goods varies as a binomial cumulative density function given γ_t at trial t . (Right) Relationship between G_t and γ_t . G_t is high when participants believe that another player is less likely to free-ride (e.g. $\gamma_t=.3$) compared to when they believe that another player is more likely to free-ride (e.g. $\gamma_t=.7$, dotted line). G_t is also discounted by the number of remaining interactions.

Comment 15: Figure 3B: x-axis show be labeled “low” and “high”. It would be better to show the value of theta here, not the product of theta * RPE

Answer to the comment 15:

This result is no longer in the revised manuscript. Please see our response to the comment 11 for explanation.

Reviewer #2 (Remarks to the Author):

Comment 1: My primary concern is about decision-algorithm of other agents. In strategic interactions, scanned participants' behavioral pattern would highly depend on the other agents' behavioral pattern. The authors have claimed "The computer agent was programmed ... in an ecological manner."; however I could not find any justification or validation. That's a critical point. I believe they need to conduct an additional behavioral experiment involving real interactions among human participants, and to show that their main computational model provides the better fit to the data compared with other models do.

Answer to the comment 1:

We agree with the reviewer that it is an important issue, especially in a behavioral economics study. However, the goal of this study was not only to investigate behavior but also its neural basis. When it comes to neuroimaging studies, it is critical that the brain responses of all participants are modulated by a specific range of computational variables. If the behavior is uncontrolled but is acquired while participants in the scanner are facing real humans, then there is a very high chance that the participants will interact with individuals having different behaviors (e.g. perseverating in a contribution or free-riding decision). To rule out these effects, behavioral studies often have much larger samples than neuroimaging studies. In a neuroimaging study such as ours, to control the variance across participants, it is necessary to test participants under controlled conditions, as is common in studies investigating strategic decisions with fMRI (e.g.^{5,6}). Moreover, a recent study reported no difference in both behavior and brain activity while interactions with real humans were compared to interactions with computer agents requiring strategic decisions¹¹.

Regarding the question about the decision algorithm of other agents, as we described in the supplementary information (Eq. S1, see below), the decision of the computer agent was determined in a model-free way by integrating a subject's previous decision (C_{t-1}^i), the proportion of contributors among others in the previous round (\bar{C}_{t-1}^{N-1}), the number of remaining trials ($T - t + 1$), and the ratio of the minimum number of contributors to make public goods (K) as below:

$$\text{logit}(\bar{C}_t^{N-1}) = \beta C_{t-1}^i + (1 - \beta) \left\{ \left(\frac{1 - K^{T-t+1}}{1 - K} \right) \bar{C}_{t-1}^{N-1} - K \right\}$$

where β indicates to what extent information influences the decision of the next trial. More importantly, we set a value of β as $0.15 \leq \beta \leq 0.35$ which was estimated by fitting this model to the actual behaviors of human subjects who had been playing the same PGG in a previous study from the first author³. This procedure helped the decision made by a computer mimic the behavior of real humans. We now explain this process in detail in the revised supplementary methods (see below). More importantly, we addressed this point in the methods section in the revised manuscript on page 3 and 14, paragraph 'Decisions of other members of the group' in the methods section:

Results (p.3)

To control the underlying motivations of other individuals across participants while creating plausible behavior in social interactions, decisions of other members of the group were generated by a computer program, unbeknownst to the participants. In neuroimaging studies, it is critical that the brain responses of all participants are modulated by a specific range of controlled computational variables. If the behavior is uncontrolled but is acquired while facing real humans, then there is a very high chance that the participants could interact with individuals having different motives underlying decisions (See decisions of other members of the group in Methods).

Decisions of other members of the group (p.14 in Methods)

To control the underlying motivations of other individuals across participants while creating plausible behavior in social interactions, unbeknownst to the participants, decisions of other members of the group were determined by a computer algorithm. First, the probability that a computer agent contributed in the first round was determined by the proportion of contribution decisions made by each of the participants during the control condition of PGG in which no feedback was given. That is, the computer agent contributed as much as the participant did. Second, the decision of the computer agent was determined by their and others' decision (C_{t-1}^i and C_{t-1}^{N-1} , respectively) in the previous round, the decision threshold (k), and the number of remaining interactions ($T - t + 1$ where t is the current trial) with a weight (β), which determined to what extent the agents change their decision according to the decision of others, or stay with their previous decision. Third, β was determined by the value which gave the maximum likelihood while predicting the actual decisions made during real human interactions in a previous study³. As a result, the computer agent tended to stay with their previous decisions, he/she was more likely to contribute in a more cooperative group, and he/she was

more likely to free-ride when failing to generate the public goods in a less cooperative group. At last, with the post-scanning questionnaire, we ensured that we analyzed the data acquired from the participants who had believed that they had interacted with real human participants simultaneously (see the supplementary methods for social contextualization score). The details of the computer algorithm generating the decision of others are described in the supplementary methods.

The decision of others (p.12 in Supplementary Methods)

The decision of others was determined by the following function given the size of the group, N , the number of remaining interactions, $T-t+1$, the minimum ratio of contributors to generate public goods, K ($K=k/N$), the previous decision of participants (C_{t-1}^i ; it was 1 if they made a contribution), and the proportion of contributors among others in the previous round (\bar{C}_{t-1}^{N-1}).

$$\text{logit}(\bar{C}_t^{N-1}) = \beta C_{t-1}^i + (1 - \beta) \left\{ \left(\frac{1 - K^{T-t+1}}{1 - K} \right) \bar{C}_{t-1}^{N-1} - K \right\}$$

Eq. S2

Note that this function has a parameter β which reflects to what extent others copied the decision of participants. β was selected randomly within the range $0.15 \leq \beta \leq 0.35$. Moreover, the effects of successful cooperation decay with the number of remaining interactions ($T-t+1$). In the first round ($t=1$), others' decisions were determined by the contribution rate of the participant, which was measured while she made no-feedback PGGs (because the computer did not have a previous decision history of the participant).

The computer agent was programmed to interact with the participants' decisions themselves in an ecological manner. This has been used in other studies of social interactions, and it allowed us to ensure that every participant played against agents whose decision was based on the same algorithm⁵. In doing so, participants were more likely to interact with cooperative fellow members when they contributed their resources in the previous round and after the group successfully generated a public good.

Comment 2: My secondary concern is about computational models. The main model looks plausible, but I believe this should be compared with many other models. For example, they can construct a model with $\Lambda = 0$, $\Pi = 0$, a mixed strategy model

($P(C) = p$ where p is a free-parameter), the optimal mixed strategy model ($P(C) = p^*$ where p^* is the optimal probability predicted by the Nash equilibrium) and a model relying only on Group-utility model etc. Furthermore, is it possible to construct a hybrid model of social learning and inequity aversion?

Answer to the comment 2:

The individual utility model comprises the mixed model satisfying the Nash equilibrium in one-shot PGG. If we model the decision of participants based on the assumption that they think others' decisions are guided by the Nash equilibrium, then, the model should incorporate a higher level of sophistication in mentalizing. In the revised manuscript, we performed and included model comparisons for different levels of sophistication in the mentalizing process. Regarding this point, please see our answer to the next comment (comment 3). In addition, we included another model in which the decision of participants was only guided by long-term collective rewards (group utility model), as the reviewer suggested. We also agree that we can test many different models. In this study, we focused on two questions: (1) whether human participants use a mental model to make a strategic decision rather than model-free decisions such as those driven by inequity aversion, and (2) to incorporate this model-based decision, whether the brain computes both immediate and long-term expected rewards, which enables the decisions to be more strategic in repetitive interactions. We have added a results section on p 4 to report the results of the group utility model suggested by the reviewer:

(p.5 in Results)

*Compared to the social learning model that incorporates the importance of future expected utility, the agent of the myopic model only considers immediate rewards when making decisions. Third, the forward-looking model assumed that participants only take long-term collective utility for groups into account (**Equation 14**).*

(p.20 in Methods)

The second alternative model is the 'group utility model' which can be distinguished from the 'social learning' model because it only takes G_t into account. In doing so, this model assumed that participants are more likely to contribute in the group where they can expect high mutual contribution. In this model, the probability of making a contribution depends on the decision value V_t which includes 4 free-parameters including χ as the weight assigned to the group utility and ζ as the initial bias (error term).

$$p(C_t) = \text{logit}(V_t) = \text{logit}(\zeta + \chi G_t)$$

(14)

Comment 3: The authors have assumed participants updated their belief about others' decision in a model-free manner. I believe they need to provide justification. As far as I know, many researchers believe that model-based learning is required to predict others' behavior in strategic interactions (e.g., Yoshida et al., 2010).

Answer to the comment 3:

Thank you for raising this very interesting and important issue. We tested the hypothesis that participants might have used iterative reasoning, forming higher-order beliefs on other people's beliefs. In the social learning model, we assumed that participants track the probability, Γ_t^{N-k} which indicates the probability in which $N - k$ people among others ($N - 1$) will free-ride at the next trial t to estimate the utility of one's contribution. When $N-k$ people free-ride, the group can benefit from public goods only when the participant contributes. Therefore, one's decision plays a pivotal role and participants expect a higher utility of a contribution decision when Γ_t^{N-k} is high. Moreover, this belief (Γ_t^{N-k}) is updated in every trial based on one's belief about the probability that another member will free-ride, γ_t . If participants use the 2nd order belief model, then, they think that another player will use the 1st order belief model for free-riding with probability $\gamma_t^{1st} = \Gamma_t^{N-k}$ because these players who used the 1st belief model think that others will free-ride with probability, γ_t . Therefore, the decision of players who use the 2nd order belief model is predicted based on their belief about the decision of another as Γ_t^{N-k} instead of γ_t . Likewise, if the participants use the 3rd order belief model, they think that another player will free-ride with probability, γ_t^{2nd} given the assumption that another player thinks that others will free-ride with probability $\gamma_t^{1st} = \Gamma_t^{N-k}$.

To examine whether higher-order beliefs explain participants' behavior better, we compared the predictability of models assuming different levels of iterative reasoning. Specifically, we replaced γ_t in the social learning model (Equation 12) with γ_t^{1st} for the 2nd order belief model and with γ_t^{2nd} for the 3rd order belief model where γ_t^{1st} and γ_t^{2nd} are computed as follows:

$$\gamma_t^{1st} = \Gamma_t^{N-k} = \binom{N-1}{N-k} \gamma_t^{N-k} (1 - \gamma_t)^{k-1}$$

$$\gamma_t^{2nd} = \binom{N-1}{N-k} \Gamma_t^{N-k} (1 - \Gamma_t^{N-k})^{k-1}$$

We further compared their predictabilities of contribution decisions with that of the 1st order belief model. We found that higher-order reasoning was inferior to the 1st-order belief model (social learning model) at explaining decisions for the current version of the PGG. This procedure is now included in the revised manuscript on p 5 and p 17 (see below):

Testing different levels of iterative reasoning (p.6 in Results)

*To explore if higher-order beliefs (e.g. belief of other people on the belief of other people) explain participants' behavior better, we compared the likelihoods of the social learning model while modulating the level of iterative reasoning. Previous findings have suggested that participants might adopt iterative reasoning, forming higher-order beliefs on other people's beliefs when predicting others' behavior in strategic interactions^{6,16,17}. To address the influences of iterative reasoning on decision-making during the PGG, we tested alternative social learning models in which the individual utility and the group utility are updated based on one's belief using 2nd and the 3rd order belief reasoning (**Equation 11**). We further compared their predictabilities of contribution decisions with that of the 1st order belief model. We found that higher-order reasoning did not explain decisions better than the 1st order belief model (social learning model) for the current version of the PGG (**Figure S2**). The fact that participants, in the current study, were less likely to use higher order reasoning may be due to the feedback provided, to the finite number of interactions with the same partners and/or to the fact that we did not explicitly show the decision of each player but only the proportion of contributors in the group. Notably, this setup mimics many ecologically relevant group decision-making situations.*

Testing higher-order beliefs on other people's beliefs (p.19 in Methods)

Last, we tested the hypothesis that participants might have used iterative reasoning, forming higher-order beliefs on other people's beliefs. If participants use the 2nd order belief model, then, they think that another player will use the 1st order belief model for free-riding with probability $\gamma_t^{1st} = \Gamma_t^{N-k}$ because these players who used the 1st belief model think that others will make free-riding with probability, γ_t . Therefore, the decision of players who use the 2nd order belief model is predicted based on their belief about the decision of another as Γ_t^{N-k} instead of γ_t . Likewise, if the participants use the 3rd order belief model, they think that another player will free-ride with probability, γ_t^{2nd} given the assumption that another player thinks that others will

free-ride with probability $\gamma_t^{1st} = \Gamma_t^{N-k}$. To examine whether higher-order beliefs explain participants' behavior better, we compared the predictability of the models assuming different levels of iterative reasoning. Specifically, we replaced γ_t in the social learning model (**Equation 11**) with γ_t^{1st} for the 2nd order belief model and with γ_t^{2nd} for the 3^d order belief model where γ_t^{1st} and γ_t^{2nd} are computed as follows:

$$\gamma_t^{1st} = \Gamma_t^{N-k} = \binom{N-1}{N-k} \gamma_t^{N-k} (1 - \gamma_t)^{k-1}$$

$$\gamma_t^{2nd} = \binom{N-1}{N-k} \Gamma_t^{N-k} \gamma_t^{N-k} (1 - \Gamma_t^{N-k})^{k-1}$$

(11)

Figure S4. The changes in quality of model fits ($-2 \log$ likelihood) resulting from changing the level of iterative reasoning in the social learning model. The 1st order beliefs model explains the decisions better than those of other higher order beliefs models.

Comment 4: In the model comparison procedure (see P.5), why did the authors calculate BIC based on cross-validated likelihood? That's strange. If they employed cross-validation to compute likelihood, they can compare likelihood (not BIC). If they employed BIC, likelihood should be derived without cross-validation. I believe cross-validation is not valid in this study, as each data (i.e., trial) in the repeated-game experiment is not independent from one another. I would recommend Bayesian model selection (Stephan et al., 2009) or hierarchical modeling approach (Daw, 2009: <http://www.princeton.edu/~ndaw/d10.pdf>).

Answer to the comment 4:

We have now recomputed BIC as the reviewer suggested and we also provided the results of Bayesian model selection (BMS). Please see our answer to reviewer1's comment 6, responding to an identical point.

Comment 5: In the fMRI analyses, why did the authors focus on feedback phase, not decision phase? What happens if they look into neural correlates of the key computational variables in decision phase?

Answer to the comment 5:

Thank you for raising this issue. We discussed this in detail in our response to reviewer 1's comment 9.

Comment 6: In the fMRI analyses, why did the authors have four separate GLMs? Is there specific reason? In principle, I believe all the computational variables of interest should be included into one single GLM to evaluate the explained-variance of each variable.

Answer to the comment 6:

We agree that it is an important issue. In the revised manuscript, we tested this by allowing regressors to compete with each other. Please see our answer to reviewer 1's, comment 10, which details how we dealt with the multicollinearity issue in the revised manuscript.

Comment 7: The present study have shown that key computational variables in the decision-making were integrated in dmPFC. To my knowledge, it is still controversial in which brain regions multiple computational variables are integrated for value-based decision-making. Some studies supported the possibility that value integration occurs in dmPFC including dACC (Hare et al., 2011; and Suzuki et al., 2015), while others provided the evidence for value integration in vmPFC (Behrens et al., 2008; Hare et al., 2010; Smith et al., 2014; and Lim et al., 2013; Suzuki et al., 2017). It would be interesting to discuss this issue in the Discussion section.

Answer to the comment 7:

Providing a better definition of strategic decision-making helped to discuss this issue compared to the previous finding which considered that all decisions were made strategically.

Since we performed a new analysis based on a narrower definition of strategic decision (GLM3) and PPI, the previous results reporting dmPFC activity are no longer in the revised manuscript. The results of the new analysis indicate that the ACC is involved in the decision to switch strategy and that its activity is modulated by the vmPFC, encoding the individual utility (the immediate expected utility in the strategic decision). We believe that this finding helps us to understand how the ACC serves to guide a strategic decision. Please see our answer to the reviewer1's, comment 12, for more detailed information on this new analyses and results.

Comment 8: Why was the number of interaction fixed (T = 15)? A conventional way in this type of repeated-game experiments is that the number of interaction is determined stochastically. The concept of “backward induction” in Game Theory predicts that participants do not cooperate in any trials (not only the last trials!) in this type of finite repeated interactions.

Answer to the comment 8:

In the revision, we included decisions in every trial including the last trial to analyze the data. Please see also our response to reviewer 1's comment 2.

Comment 9: In Figure 4A, the activation labeled ACCg is corpus callosum?

Answer to the comment 9:

There seems to have been a misunderstanding. The activation was not only including the corpus callosum, but also the ACCg. However, since we performed a newly defined PPI, the subsequent results including Figure 4 are no longer in the revised manuscript, and therefore do not need to be discussed.

Comment 10: The authors said they recruited N strangers for each experiment. How did they confirm the participants are strangers?

Answer to the comment 10:

We are sorry if there was a misunderstanding in the sentence that we wrote, 'N strangers made collective decisions together as a group. We kept the number of members in a group constant (N=5)'. Please remember that the participants were led to believe that they were

interacting with 4 other individuals who were strangers. We changed this sentence as below to avoid confusion:

Methods part (p.13)

In the PGG used in the current study, a participant was led to believe that he made decisions within a group of 5 members (N=5). The participants were told that they would play with 19 other participants located in another room, so that 20 participants in total would play the PGG in 4 different groups of 5 subjects simultaneously, randomly arranged by a computer at the beginning of every PGG. Hence, participants lying in the scanner knew that they were interacting with a different combination of group members at the beginning of every PGG.

Comment 11: As far as I understand, when decision-making participants could see information about k, t and T-t. How about including these variables into GLM as regressors of no-interest?

Answer to the comment 11:

As the reviewer suggested, we inputted the current trial (t) as regressors of non-interest in the GLM. Please see our answer to reviewer 1's comment 10 for details.

Reviewer #3 (Remarks to the Author):

Comment 1. Authors need to present the results of behavioral analyses as well as how well computational model fits the main aspects of participants' behavior. As an example of the most fundamental analyses, authors could use logistic regression model and see how participants' choice (contribution vs. free-ride) at a given round was influenced by relevant variables, such as number of free-riders (among other people; nF) and participant's choice in the past few trials, trial history of reward, success/failure of producing public good, interaction of reward and choice (i.e. win-stay-lose-switch), etc. to name a few. To see if "N-k" (among N-1) is the critical value that determines participants' choice, separate regressors for $nF=N-k$, $nF<N-k$, $nF>n-K$. If participants' behavior is consistent with the model prediction, regressor for $nF=N-k$ should have positive coefficients (i.e. participants tend to contribute when $nF=N-k$ in the previous trials compared to other nFs), consistent with the modeled effect of individual utility. On the other hand, the regressor for $nF<N-k$ would provide evidence for the effect of modeled group utility. If group utility is determining factor of choice, then the regressor for $nF<N-k$ should have positive coefficients (i.e. the more other people contributes, the more participant contributes), while negative coefficients indicate stronger effect of individual utility (i.e. the more other people contribute, the less participant contributes). This is only one example. Authors need to provide strong evidence that participants' behavior was consistent with model predictions.

Answer to the comment 1:

The reviewer suggested two regression analyses. The first is related to testing the hypothesis of whether participants used model-free decision-making. In the revised manuscript, we included the model-free analysis as the reviewer suggested.

The second is to test the effects of the trial in which the number of free-riders among others was $N-k$ ($nF = N - k$) in subsequent decision-making compared to the other trials. The reviewer suggested this latter analysis to test our assumption underlying the individual utility (I_t). We performed this analysis but we did not find that the regression coefficient on $nF = N - k$ was higher than the others: mean $\beta^{nF=N-k} = -0.53 \pm 6.13$ while mean $\beta^{nF>N-k} = -13.58 \pm 7.89$; mean $\beta^{nF<N-k} = 6.93 \pm 8.26$ (all β was not significant (n.s.) in $p < 0.05$). Below, we discuss why we think this analysis was not appropriate to test the definition of individual utility (I_t). We have tested this regression not only for the behavioral data made by participants but also for the synthetic

data which were generated by the social learning model. We found a similar pattern ($\beta^{nF < N-k} > \beta^{nF = N-k} > \beta^{nF > N-k}$) with regression coefficients in the data generated by a computer agent: mean $\beta^{nF = N-k} = -12.70 \pm 9.55$ while mean $\beta^{nF > N-k} = -21.67 \pm 10.55$; mean $\beta^{nF < N-k} = 21.08 \pm 10.46$. This result shows that there is no guarantee that we should find higher $\beta^{nF = N-k}$ even if the decisions were actually driven by the social learning model which incorporate the individual utility.

We believe that the proposed analysis is only able to test an oversimplified version of the social learning model for the following reasons. First, we did not define the individual utility (I_t) as dependent on whether the number of the free-rider (nF) in the previous trial was $N - k$ or not. Instead, we defined it as one's belief about the probability that there will be $N - k$ free-riders (e.g. participants did not necessarily believe that $N-k$ will freeride after they have experienced $N-k$ free-riders in the previous interaction). Therefore, it follows a function of the updated belief, γ_t , which varies not only according to nF but also depends on each participant's learning rate and their previous reward prediction error (related to α and θ). Second, I_t was not defined as a deterministic but as a continuous function of γ_t . Moreover, I_t also varied according to the extent to which a participant was willing to contribute to others' payoff (π), which varies across individuals. Last, more importantly, the decision is also guided by the group utility (G_t). Unlike I_t , the term G_t allows us to capture the changes in motive underlying contribution in the progress of Public goods game by incorporating the remaining interactions. Considering the points noted above, we think that the proposed regression has not taken those complexities into account.

In fact, we provide an additional result proving that the data purely generated by the social learning model also can afford similar characteristics of the model-free regression that we found in the actual behavioral data. To test this, we generated the synthetic data set with cross-validation using the social learning model and performed the same model-free regression analysis. We show that the model-based prediction outperformed the model-free decision predictions using Bayesian model selection. We have provided the details of this analysis in our response to reviewer 1's comment 7.

Comment 2.1. Although social-game paradigm provides an opportunity to study the strategic decision (i.e. iterative reasoning, higher-order beliefs on other's beliefs, etc.), it is unclear what are the strategic components of participants' behavior in this study. First, group utility in social learning model might capture it. However, the estimated weight ($w \sim 0.8$) for individual utility is very high, suggesting that individual utility was major determinant of the choice. The BIC difference between social learning model and myopic model seems to be only marginal. In addition, it is not clear whether the model-term

group utility was capturing only the temporal decay of contribution tendency over time (i.e. $1-K^{T-t+1}/1-K$), not the effect of successful group cooperation (i.e. probability that $>k$ people would contribute).

Answer to the comment 2.1:

We have realized that we previously included by mistake the parameters estimated of participants excluded (due to the excessive head movements) and their responses in the post-scanning questionnaire (attached in supplementary methods) when averaging the parameters reported in the previous Table 1. Those participants include ones who might not believe that other players made a simultaneous decision with them in post-scanning questionnaire. The ω of those participants increased the overall value of ω .

Moreover, in the revised manuscript, we estimated the parameters to fit the decisions made by participants in all trials rather than to the decisions of trials except for the last trial (see our response to reviewer 2's comment 8).

By correcting these errors and after including all decisions, we found that the influence of group utility ($1-\omega$) over that of individual utility ($\omega=0.65\pm 0.07$) was not minimal and that the overall quality of fit was greatly improved. Following other reviewers' suggestion, we computed the BIC as the sum of all trials, not as the average per trial: the BIC in social learning model was -6149 compared to the myopic model for which the BIC was -5721 (see the updated Table 1 below). The model comparison was tested rigorously with Bayesian model selection (see our response to reviewer 1's comment 6 for details).

Last, we included the current trial t as regressors of no-interest in the neuroimaging analysis to rule out the possibility that G_t only captures the temporal decay and not the effect of successful group cooperation. This confirmed that the lateral frontopolar cortex encoded the trial by trial G_t during PGG (See our response to reviewer 1's comment 10 for details). Taken together, these results reported in the revised manuscript support the interpretation that the group utility (G_t) plays a critical role in predicting strategic decision-making.

Importantly, as noted previously, we have now adopted a clearer definition of strategic decision-making. In the volunteer's dilemma, saving one's endowed money by choosing not to contribute to the group also can be considered as a strategic decision when one expects that there will be enough contributors to generate the public goods, as well as contribution to foster a cooperative environment. By incorporating this perspective, we defined the decision to switch one's strategy from the previous one as driven by strategic decision. This is exactly what the reviewer proposed us to consider in his/her comment 5. Taking this suggestion into account, we

performed a new analysis to examine the neural underpinning of switching between strategies. Please see our answer to the comment 2.3 below.

	ω	π	λ	α	θ	BIC
Social learning	.65±.07	.16±.16	-2.10±.45	.51±.06	.13±.02	-6149
Myopic	11.04±2.76	.34±.19	-2.98±1.09	.59±.06	.14±.04	-5721
	χ	ζ	α	θ		
Group utility	9.30±4.55	-30.91±4.92	.56±.06	.12±.02		-5713
	δ	ε	κ			
Inequity aversion	16.39±7.11	.61±.42	1.37±1.00			-4220

Table 1

Comment 2.2. Second, participants might have used iterative reasoning, forming higher-order beliefs on other people’s beliefs. It may be helpful for the authors to explore if higher-order beliefs (e.g. belief of other people on the belief of other people) can partially explain participants’ behavior.

Answer to the comment 2.2:

To address this issue, we tested for the existence of higher order beliefs. We have responded to this point in detail in our response to the reviewer2’s comment 3.

Comment 2.3. Finally, previous studies have shown that dmPFC region can be involved in strategic reasoning or switching/arbitration between different strategies (e.g. Hampton et al., 2008; Seo et al., 2014). It would be helpful, if authors can more clearly describe what are the novel aspects of "strategic decision" that could be studied in public good game in groups, and what are the novel insights readers can gain from this study about the function of dmPFC in strategic decision-making.

Answer to the comment 2.3:

We believe that the lack of clarity that the reviewer raised here was caused by our poor definition of strategic decisions. We have now addressed this issue by introducing a clearer definition, adopting the reviewer’s suggestion from comment 5. That is, we defined strategic-decisions in this study as an arbitration between strategies. According to this new definition, we identified brain areas that are specific to switching strategy with the GLM3. Furthermore, we

performed a functional connectivity analysis using the time series extracted from the seed regions from this GLM3 – the ACC and vIPFC – inputted as a physiological factor. Importantly, as psychological variables, we estimated to what extent the model predicted value of contribution decision is changed from the previous trial ($\Delta Q = Q_t - Q_{t-1}$). Therefore, the psychological factor includes the probability of changing one's strategy from free-riding to contribution (positive values) and the probability of changing one's strategy from contribution to free-riding (negative values). Finally, we identified the brain areas encoding such psycho-physiological interactions. For the details of this GLM3 and the new PPI analysis, please see our response to reviewer 1's comment 12. Note that we are no longer discussing dmPFC activity since the new PPI analysis identified the vmPFC and frontopolar cortex as key nodes.

Comment 3. In the analyses of fMRI data, authors used separate GLM models to look for activations correlated with different sets of decision-variables (e.g. individual/group utilities-GLM1, prediction errors for the belief on others' probability of free-riding (PFR) – GLM 2, PFR itself – GLM 3, estimated choice probability – GLM 4). However, these variables tested in separate models are not necessarily independent of each other. For example, in the social learning model, PFR (GLM 3) and prediction error for PFR (GLM 2) are linearly correlated. Choice probability (GLM 4) is also correlated with individual and group utilities (GLM 1). Therefore, in order to know the effect of each variable independent of other correlated variables, all the co-linear variables need to be included in the same regression model. Otherwise, the significant effect of one variable tested in one model could actually reflect the effect of other correlated variables.

Answer to the comment 3:

We corrected the issue of multicollinearity by introducing a new GLM. We have responded to this point in our responses to reviewer 1's comments 10 and 11.

In addition, the reviewer pointed out that the activation correlating with choice probability could be simply related to the execution of the final decision (in comment 5). The reviewer suggested to examine the neural underpinnings of strategic decision-making, i.e. the activation related to the arbitration between two strategies. Following this suggestion, we have constructed GLM3, which identified the brain regions specific to trials switching strategy compared to trials keeping the previous strategy. This new analysis replaces the previous GLM searching for brain areas showing correlation with the model predicted decision probability. The design of the

GLM3 and corresponding results are now shown in the revised manuscript. We have responded to this point in our responses to reviewer 1's comments 12:

Comment 4. In general, the description of GLM models in method section is lack of clarity. Particularly, it is unclear how the PPI analysis was done. Authors need to provide better description of what are the “psychological” variables that modulate the functional connectivity. Including equations would help.

Answer to the comment 4:

Please see our response to comment 2.3 which clarifies the GLM models and also details the new PPI analysis.

Comment 5. Authors argued that dmPFC might be involved in strategic decision, as its activation was correlated with choice probability estimated by the model. However, it is not clear how the activation correlated with choice probability can be the evidence for strategic decision. If the decision based on group utility is a “strategic choice” as opposed to a choice based on individual utility, then the region whose activation was correlated with group utility could be involved in “planning” strategic choice or switching between strategies. The activation correlated with choice probability could be simply related to the execution of final decision. Activation related to the “arbitration” between two strategies (individual vs. group utility) can be also related to strategic choice (Lee et al., 2014). This issue is related to the comment #2. It would be helpful if authors provide clear conceptual framework for what “strategic” choice means in the public good game, as well as valid quantitative measurement of strategic choice.

Answer to the comment 5:

This issue is associated with comment 2. We have addressed this in our responses to reviewer 3, comment 2.1 and 2.3.

References

1. Chung, D., Yun, K. & Jeong, J. Decoding covert motivations of free riding and cooperation from multi-feature pattern analysis of EEG signals. *Soc. Cogn. Affect. Neurosci.* **10**, nsv006- (2015).
2. Hampton, A. N., Bossaerts, P. & O'Doherty, J. P. Neural correlates of mentalizing-related computations during strategic interactions in humans. *Proc. Natl. Acad. Sci. U. S. A.* **105**, 6741–6746 (2008).
3. Park, S. A., Jeong, S. & Jeong, J. TV programs that denounce unfair advantage impact women's sensitivity to defection in the public goods game. *Soc. Neurosci.* **8**, 568–582 (2013).
4. Rilling, J. K. *et al.* A neural basis for social cooperation. *Neuron* **35**, 395–405 (2002).
5. Suzuki, S., Niki, K., Fujisaki, S. & Akiyama, E. Neural basis of conditional cooperation. *Soc. Cogn. Affect. Neurosci.* **6**, 338–347 (2011).
6. Yoshida, W., Seymour, B., Friston, K. J. & Dolan, R. J. Neural Mechanisms of Belief Inference during Cooperative Games. *J. Neurosci.* **30**, 10744–51 (2010).
7. Suzuki, S. *et al.* Learning to Simulate Others' Decisions. *Neuron* **74**, 1125–1137 (2012).
8. Seo, H., Cai, X., Donahue, C. H. & Lee, D. Neural correlates of strategic reasoning during competitive games. *Science (80-.).* **346**, 340–343 (2014).
9. Wan Lee, S., Shimojo, S. & O'Doherty, J. P. Neural Computations Underlying Arbitration between Model-Based and Model-free Learning. *Neuron* **81**, 687–699 (2014).
10. Neubert, F.-X., Mars, R. B., Sallet, J. & Rushworth, M. F. S. Connectivity reveals relationship of brain areas for reward-guided learning and decision making in human and monkey frontal cortex. *Proc. Natl. Acad. Sci. U. S. A.* **112**, 1–10 (2015).
11. Suzuki, S., Adachi, R., Dunne, S., Bossaerts, P. & O'Doherty, J. P. Neural mechanisms underlying human consensus decision-making. *Neuron* **86**, 591–602 (2015).
12. Wan, X., Cheng, K. & Tanaka, K. Neural encoding of opposing strategy values in anterior and posterior cingulate cortex. *Nat. Neurosci.* **18**, 752–9 (2015).
13. Kolling, N., Scholl, J., Chekroud, A., Trier, H. A. & Rushworth, M. F. S. Prospection, Perseverance, and Insight in Sequential Behavior. *Neuron* **99**, 1069-1082.e7 (2018).
14. Khamassi, M., Quilodran, R., Enel, P., Dominey, P. F. & Procyk, E. Behavioral regulation and the modulation of information coding in the lateral prefrontal and cingulate cortex. *Cereb. Cortex* **25**, 3197–3218 (2015).
15. Benjamini, Y. & Hochberg, Y. On the Adaptive Control of the False Discovery Rate in

- Multiple Testing with Independent Statistics. *J. Educ. Behav. Stat.* **25**, 60 (2000).
16. Devaine, M., Hollard, G. & Daunizeau, J. The Social Bayesian Brain: Does Mentalizing Make a Difference When We Learn? *PLoS Comput. Biol.* **10**, e1003992 (2014).
 17. Coricelli, G. & Nagel, R. Neural correlates of depth of strategic reasoning in medial prefrontal cortex. *Proc. Natl. Acad. Sci. U. S. A.* **106**, 9163–9168 (2009).

Reviewers' Comments:

Reviewer #1:

Remarks to the Author:

I appreciate the tremendous amount of work that the authors have put into this revision to address my comments and those of other reviewers. This has improved the manuscript substantially. In particular I liked the new definition of strategic reasoning with the differentiation into immediate individual utility and long-term group utility and the associate switches to contribution or free-riding. I also appreciated the posterior predictive check that the authors included in Figure S1.

However, one short coming of the presentation of this ppc analysis, is that we only get to see the average trial-by-trial decisions of all subjects. Such average behavior is easy to simulate, but it would be more convincingly to demonstrate that the social learning model also predicts individual behavior. In addition, the time points (trials) of the above-mentioned individual switches to contribution or free-riding should be an essential part of the posterior predictive check. Can the model simulate data, that also exhibits switches to contribution or free-riding at roughly the same time-points? Such an analysis should be included as an additional figure in the supplement.

Reviewer #2:

Remarks to the Author:

The authors have adequately addressed my concerns.

Reviewer #3:

Remarks to the Author:

In this revised manuscript, authors addressed reviewer's concerns with additional analyses and results. Nevertheless, some concerns still remains mostly regarding authors' analysis of strategic decision and the interpretation of the result.

Authors argue that participants tended to switch to free-riding decision in favor of individual utility whereas they switch to contribution decision in favor of group utility.

First, it is not clear from the social learning model (equation 5 and 8) that this is necessarily true, when Q is a weighted sum of individual and group utility. In other words, the sign of ΔQ may not necessarily reflect increase or decrease of individual over group utility. Authors need to explicitly show how ΔQ is related to the trial-by-trial changes in individual and group utility. It is also unclear (not quantitatively shown) whether ΔQ had significant influence on choice switch.

Second, it is unclear how ΔQ can reflect "arbitration". It can simply reflect change in choice tendency as a combined function of individual and group utility.

Third, the issue of potential confounding that was raised for GLM 1 and GLM 2 still remains for GLM 3.

Reviewer #1 (Remarks to the Author):

I appreciate the tremendous amount of work that the authors have put into this revision to address my comments and those of other reviewers. This has improved the manuscript substantially. In particular I liked the new definition of strategic reasoning with the differentiation into immediate individual utility and long-term group utility and the associate switches to contribution or free-riding. I also appreciated the posterior predictive check that the authors included in Figure S1.

However, one short coming of the presentation of this ppc analysis, is that we only get to see the average trial-by-trial decisions of all subjects. Such average behavior is easy to simulate, but it would be more convincingly to demonstrate that the social learning model also predicts individual behavior. In addition, the time points (trials) of the above-mentioned individual switches to contribution or free-riding should be an essential part of the posterior predictive check. Can the model simulate data, that also exhibits switches to contribution or free-riding at roughly the same time-points? Such an analysis should be included as an additional figure in the supplement.

Thank you for the positive feedback. We agree with the reviewer that, in addition to its ability to predict the contribution/ free-riding decision, it is also important to check whether the model can provide information about individual behavior and about when to switch from the previous strategy or to stay with the current strategy. To address this point, we estimated to what extent the model predicts a stay/switch decision at trial t ($2 \leq t \leq 15$) across trials. We found that in the model-based simulation with cross-validation (as shown in Fig S1), the same switch is made 36.6 ± 0.08 % of the cases within the next trial. Note that accurate prediction in a switch decision indicates that the model not only accurately predicts the current decision but also the decision in the previous trial (conditional probability, $p(\text{Contribution} | \text{Free-riding})$ and $p(\text{Free-riding} | \text{Contribution})$). It suggests that the chance level prediction of switching decision is not 50%. Moreover, in repetitive social interactions, the probability to make a contribution decision is continuously influenced by the previous interactions. Considering that, we computed the baseline prediction of switching strategy with number-matched simulation and compared it to that of the model-based simulation. We found that the model-based simulation predicts the same switch more accurately than the number-matched simulation ($t_{24}=2.56$, $p=0.017$, paired t-test; **Fig. S1 D**). To compute the baseline predictability from the number-matched simulation, we generated 100 sets of randomly shuffled data for each participant while matching the number of the contribution decision with those of the model-based simulation during each of the 12 blocks of PGG, and we used the number of accurate predictions of stay/switch decisions from this number matched simulation as the baseline prediction.

Among 168 trials of decisions, we found that the model-based simulated decision accurately predicted the switch/stay decision in 131.9 ± 5.1 trials across participants, which is significantly above the number matched baseline simulation (107.2 ± 2.8 trial; $t_{24}=4.99$, $p=4.27e-05$; left panel in **Fig. S1 E**). We also reported the number of missed prediction according to the following categories: 1) the number of switch decisions while the model-based simulation made a stay decision (29.7 ± 4.6 trial in model simulate data $< 33.8 \pm 3.1$ trials in number matched simulation); 2) the number of stay decisions while the model-based simulation made a switch decision ($0.9 \pm 0.3 < 5.2 \pm 0.7$); 3) the number of switch decisions that the model-based simulation switches in different direction (e.g. switching to the contribution decision while the model-based simulation made a switch to the free-riding decision) ($0.9 \pm 0.3 < 5.2 \pm 0.7$) (right panel in **Fig. S1 E**). These results are now added in the model validation in the results section (on p. 6) and Figure S1 D (p.3 in the supplementary information) as indicated below.

Model validation and comparison in p.6

... When a participant switches from his previous decision, the model-based simulation also predicts the same change that participants made (switch to contribution or switch to free-riding) in $36.8 \pm 0.08\%$ accuracy within the next trial. To test how well these decisions in the model-based simulation reflect the actual sequence of switch/stay decisions made by a participant, we compared the model-based simulation to a control number-matched simulation. To do this, we generated 100 sets of randomly shuffled data for each participant while matching the number of the contribution/free-riding decisions with those of the model-based simulation during each of the 12 blocks of PGG, and we compared this baseline prediction of stay/switch decisions to that of the model-based simulation. We found that the model-based simulation better predicts the same switch decision than the number-matched simulation within the next trial ($t_{24}=2.56$, $p=0.017$, paired t-test; **Fig. S1 D**). In the model-based simulation, the decisions of each participant about whether to switch to contribution or free-riding or whether to stay to the previous strategy were accurately predicted for 131.9 ± 5.1 (s.e.m.) trials among 168 in total (except for the first decision in each block of PGG). We found that the agent in the model-based simulation is more likely to make the same switch/stay decision on the same trial of the actual decision made by a real participant, and this effect was significantly above the baseline ($t_{24}=4.99$, $p<0.001$, paired t-test; **Fig. S1 E**).

Fig. S1 D. The percentage of accuracy of the model-based simulated decision in predicting the same switch decision that a participant made within the next one trial. The model-based simulated decision set used here is the same 12 folds cross-validation as shown in Fig S1 B and C. To examine how well switch decisions are predicted by the model-based simulation, we compared this with the baseline prediction of switch decisions. To compute the baseline prediction, we measured how much the randomly shuffled data predict the switch/stay decisions while matching the number of contribution/free-riding decisions of each subject in each block (100 times iterations per block). The model-based simulation more accurately predicts the switch decision than the number-matched simulation ($t_{24}=2.56$, $p=0.017$, paired t-test). **E.** Average number of trials across participants in which the decision to switch from the previous strategy or to stay with the current strategy is accurately predicted by the synthetic decisions generated by the social learning model across 168 trials (except for the decision in the first trial of each block) (Left) and number of trials in which the model-based simulation differed from the actual switch/stay decision (right). The social learning model not only generated series of switch/stay decisions that are more similar to those made by a participant than the number matched baseline (left; $t_{24}=4.99$, $p<0.001$), but had also lower miss predictions than the number matched baseline simulation (right). In the right panel, the first bar indicates the number of trials in which the model-based simulation made a stay decision when the participant made a switch decision; the second bar indicates the number of trials in which the model-based simulation made a switch decision when the participant made a stay decision; the third bar indicates the number of trials in which the model-based simulation had a switch decision in a different

direction from the direction a participant made (e.g. the model switches from the contribution to free-riding, C-F while the participant switches from free-riding to contribution, F-C). The baseline prediction estimated from the number-matched simulation is shown in dotted line. Error bars indicate s.e.m.

Reviewer #2 (Remarks to the Author):

The authors have adequately addressed my concerns.

Reviewer #3 (Remarks to the Author):

In this revised manuscript, authors addressed reviewer's concerns with additional analyses and results. Nevertheless, some concerns still remains mostly regarding authors' analysis of strategic decision and the interpretation of the result.

Authors argue that participants tended to switch to free-riding decision in favor of individual utility whereas they switch to contribution decision in favor of group utility.

First, it is not clear from the social learning model (equation 5 and 8) that this is necessarily true, when Q is a weighted sum of individual and group utility. In other words, the sign of ΔQ may not necessarily reflect increase or decrease of individual over group utility. Authors need to explicitly show how ΔQ is related to the trial-by-trial changes in individual and group utility. It is also unclear (not quantitatively shown) whether ΔQ had significant influence on choice switch.

We agree with the reviewer that it is not necessarily true that participants tend to switch to free-riding decision in favor of individual utility while they switch to contribution decision in favor of group utility. We want to clarify that we are NOT arguing that 'participants tended to switch to free-riding decision in favor of individual utility whereas they switch to contribution decision in favor of group utility'. We are afraid that the reviewer may have misunderstood our interpretation of the psychological factor (ΔQ) in the PPI analysis based on the incorrect assumption that ΔQ either reflects changes in the individual utility or changes in group utility. In fact, our analyses and interpretation of the results stand on the definition of Q as the weighted sum of the individual utility and of the group utility. Therefore, ΔQ is the index of the trial-by-trial tendency to switch one's decision to the alternative or to keep with the previous decision, and does not reflect the changes in either individual or group utility alone.

Please note that we also did not state in the manuscript that ΔQ is only driven by either individual or group utility. For example, the following sentences were extracted from our discussion (p.12): "*Moreover, the functional connectivity between the vIPFC and ACC –regions selectively engaged for the event predicting the strategy switch– decreased with the vmPFC encoding 'individual utility' and increased with the IFPC encoding 'group utility' when the probability of contribution was high at the time of feedback.*" We are concerned that this argument may be misread by the reviewer. If this is the case, we would like to clarify this point. What we described here concerned the results of two independent analyses. First, in the PPI analysis, we found that the connectivity between the vIPFC and ACC to the vmPFC and IFPC are modulated by ΔQ . We used ΔQ as the psychological factor indicating the trial-by-trial changes in tendency to switch one's decision from the previous one. Second, in an independent univariate analysis (GLM1), we found that activity in the vmPFC inversely correlated with individual utility, and that activity in the IFPC positively correlated with the group utility. Lastly, we found, in univariate analyses, that the vmPFC and IFPC encoding individual and group utility are the same areas that were found in the PPI analysis as showing changes in connectivity with the vIPFC/ACC as a function of ΔQ . We have attempted to clarify this subtle point further in the Discussion (see below).

As the reviewer mentioned, the social learning (SL) model predicts free-riding decision with decreases in Q , which can be caused not only by a decrease in the individual utility (associated with vmPFC activity increases) but also by a decrease in the group utility

(associated with IFPC activity decreases). Therefore, for a participant whose Q_{t-1} was large enough to make a contribution decision at the previous trial $t-1$, if Q_t is decreasing on the current trial t , then the activity in the brain areas which guide switching to the alternative strategy (vIPFC and ACC) may increase while ΔQ ($\Delta Q = Q_t - Q_{t-1}$) has a negative value. Our PPI analysis tested this hypothesis. Therefore, we agree that our results should not be interpreted as reflecting that the changes in functional connectivity are modulated solely by changes in individual utility or by changes in group utility. Instead, our results suggest that the switch to a free-riding decision is guided by *both* increase in vmPFC to vIPFC/ACC connectivity and decrease in the IFPC and the vIPFC/ACC connectivity. Likewise, the switch to the contribution decision is guided by *both* decrease in vmPFC to vIPFC/ACC connectivity and increase in the IFPC and the vIPFC/ACC connectivity.

To avoid potential misunderstanding, we have revised the sentences mentioned above as follows (p.12 in Discussion).

Moreover, the functional connectivity between the brain areas selectively engaged for the event predicting the strategy switch (vIPFC and ACC) decreased with the vmPFC and increased with the IFPC when the probability of contribution is increasing ($\Delta Q > 0$) at the time of feedback. This finding suggests that the brain computes the expected utility by integrating the individual utility encoded in the vmPFC and the group utility encoded in the IFPC. When the expected utility of the current strategy is lower than that of the alternative strategy, these brain regions are more likely to exhibit changes in functional interactions to the vIPFC and ACC to guide a switch between strategies.

We have also noticed the first sentence in the same paragraph (p.11 in discussion) could have led to the confusion that the reviewer raised while reading the following part of the discussion. We have therefore also changed this sentence to provide an appropriate premise.

In the current study, participants who adopted a mixed strategy might flexibly switch their strategies between free-riding and contribution according to the changes in their expected utility.

Second, it is unclear how delta Q can reflect "arbitration". It can simply reflect change in choice tendency as a combined function of individual and group utility.

The terminology “arbitration” was adopted from a previous comment made by one of the reviewers. We understand that this term can be misleading because it has previously been used in a more specific way, such as “arbitration between model-free and model-based learning”. To avoid giving the impression that we are using such specific connotation, in the revised manuscript, we are no longer using the word ‘arbitration’. We have used other terms to deliver our message more accurately.

In the manuscript, the word arbitration was used to indicate the process of strategy selection which includes the changes in tendency to choose a different strategy. Importantly, as we described in the previous manuscript (p.12 in discussion), it is hard to dissociate the changes in decision from the changes in strategy in this study: *“It is important to note that here, we defined a change in strategic-decision as a change in model predicted decision-value, because it is not possible to dissociate a change in strategy from a change in behavioral response.”* Therefore, the “arbitration between strategies” was also used to indicate the choice process between two strategies.

Below is the list of sentences in which we previously used the word “arbitration” and the changes we made consequently in *italics with underline*.

Abstract

When it is required to change one's strategy, these two brain regions exhibited changes in functional interactions with brain regions engaged in switching decision strategies (the anterior cingulate cortex (ACC) and ventrolateral prefrontal cortex (vIPFC)).

p.2 in introduction

Finally, when participant can expect better utility by choosing the alternative strategy, the ACC and the ventrolateral prefrontal cortex (vIPFC), which were engaged for switching decision strategy, showed changes in functional connectivity with the vmPFC and IFPC, regions encoding the utility of the strategy.

p.8 the title of a subsection in results

Neural mechanisms underlying switch between different strategies

p.8 in results

These areas are therefore likely to be involved in implementing a choice between strategies.

To identify the neural mechanism underlying the changes in strategies, we examined how the choice probability modulates the interactions between the brain areas involved in switching decisions and the areas involved in encoding the individual utility and the group utility. We hypothesized that brain regions implementing the switch between strategies would show enhanced coupling with those areas encoding subjective utilities.

p.9 in results

The physiological variables were the brain signals extracted from the brain areas involved in the switch between strategies (ACC and vIPFC) at the time of feedback.

That is, increased functional connectivity was found between the IFPC and the brain areas engaged in switching strategies as a function of the changes in decision value to contribution strategy, while increased functional connectivity was observed between the vmPFC and the brain areas engaged in switching strategies as a function of the changes in decision value to free-riding strategy. Together, these findings suggest that the neural encoding of the group utility and individual utility, as formalized from the social learning model, inform the alteration between strategies in vIPFC and ACC during social interactions.

p.11 in discussion

Considering strategic decision-making as the flexible shifts between different strategies, we observed that the activity from the vIPFC and the ACC increased when switching between strategies, suggesting that these brain regions signal the needs to change the current strategy during collective decisions.

p.12 in discussion

Finally, functional connectivity between areas (ACC and vIPFC) guiding switch to different strategies

p.24 in methods

the contrast images related to switching the current strategy (Switch > Stay; GLM3) were calculated and entered in a second level analysis

We define the psychological factor of strategic decisions as the trials requiring the switch in strategy.

p.25 in methods

we focused on the time of feedback and examined the changes in functional connectivity between the brain regions involved in the switch in strategies and the brain regions encoding the computational variables

the anterior cingulate cortex (ACC), (x,y,z=0,17,31) predicting the changes in strategies (GLM3).

Figure 4. title and its legend

Neural mechanisms of strategy selection during the Public goods game.

Connectivity analyses between the brain regions engaged in switching to the alternative strategy (the vIPFC and the ACC)

Third, the issue of potential confounding that was raised for GLM 1 and GLM 2 still remains for GLM 3.

We believe that the reviewer is referring to the previous potential problem noted in the previous reviews: *“running several different GLMs is just avoiding, not addressing the problem as the interpretation of different GLMs could become obscured if the inherent correlation between signals is not addressed. Therefore, it is even more important to report correlations between different regressors and model-based signals”*. We have reported that due to the inherent process of repetitive social interactions there are potential confounds in GLM1 and GLM2. To address this potential confound for GLM1 and GLM2, we have adopted the partial correlation and prioritize the brain area computing the regressors of interest over others (see our answers to the previous reviews).

Concerning GLM3, we believe there may have been some confusion since there this GLM did not include any parametric regressor, obviating the concern about collinearity. Rather, GLM3 was a factorial model. We were aware that GLM3 should be independent from the computational model to define the seed region for the PPI analysis. Specifically, with GLM3, we determined brain areas reflecting a switch of strategy (switch > stay). The following paragraph is the description of this model (p. 24): *“Last, we examined the brain regions specifically engaged when switching one’s decision strategy from one trial to another, compared with staying with one’s previous strategy (Switch>Stay). For the third GLM (GLM3), we thus compared the brain responses of the outcome at round t-1 (including the 1st to the 14th round of the PGG) to model the decision at the round t. The outcome phases of the round t-1 of the PGG were split into the ‘switch’ and the ‘stay’ trials according to the interaction between the decision at round t-1 and the decision at the following round t. GLM3 also included the same regressors of no interest defined in GLM1, which includes button press, instructions, decision onsets, missing trials and motion regressors.”*

To avoid potential misunderstanding, we added the following sentence to specify that the switch/stay decision in GLM3 was not based on the prediction from the social learning model but was based on actual decisions made by participants.

p.24 in discussion

The switch and stay trials were determined based on the decisions made by each participant, and were not based on the model-based prediction.

Reviewers' Comments:

Reviewer #1:

Remarks to the Author:

The authors have addressed all my comments. Congratulations on a well-conducted study and a well-written paper.

A final minor comment: It took me quite some time to figure out, what the italicized labels (ie.. Prediction Switch(Stay) ...) in Fig S1E actually refer to. Can this be explained in the Figure legend?

Reviewer #3:

Remarks to the Author:

concerns were properly addressed.